# Placental iron utilisation in fetal growth restriction: alterations in mitochondrial haem synthesis and iron–sulphur cluster assembly pathways

Veronica B. Botha[1,2], Heather C. Murray[1,3], Siddharth Acharya[1,2], Kirsty G. Pringle[1,4], Roger Smith[2,5] and Joshua J. Fisher[2,5]

[1]*School of Biomedical Sciences and Pharmacy, University of Newcastle, Callaghan, New South Wales, Australia*
[2]*School of Medicine and Public Health, University of Newcastle, Callaghan, New South Wales, Australia*
[3]*Precision Medicine Program, Hunter Medical Research Institute, New Lambton Heights, New South Wales, Australia*
[4]*Women's Health Research Program, Hunter Medical Research Institute, New Lambton Heights, New South Wales, Australia*
[5]*Mothers and Babies Research Program, Hunter Medical Research Institute, New Lambton Heights, New South Wales, Australia*

Handling Editors: Laura Bennet & Janna Morrison

The peer review history is available in the Supporting Information section of this article (https://doi.org/10.1113/JP289451#support-information-section).

*The Journal of Physiology* (side banner)

**Abstract figure legend** Altered iron handling and mitochondrial pathways in FGR placentas. Placental tissue from FGR pregnancies showed increased expression of iron importers, transferrin and divalent metal transporter 1, and decreased

R. Smith and J. J. Fisher share equal contribution as senior authors.
This article was first published as a preprint. Botha VB, Murray HC, Acharya S, Pringle KG, Smith R, Fisher JJ. 2025. Placental Iron Utilisation in Fetal Growth Restriction: Alterations in Mitochondrial Heme Synthesis and Iron-Sulfur Cluster Assembly Pathways. bioRxiv. https://doi.org/10.1101/2025.06.09.658195

expression of the iron exporter ferroportin, suggesting that the FGR placenta is retaining iron to meet its own utilisation. Mitochondrial expression of the iron importer mitoferrin 2 was reduced in FGR placentae, *de novo* iron–sulfur cluster [2Fe–2S] synthesis was increased and late-stage [4Fe–4S] assembly was decreased. FGR placentae expressed an increased mitochondrial and cytosolic haem synthesis pathway, accompanied by reduced haemoglobin expression and erythrocyte structural proteins. Collectively, this study suggests that FGR mitochondria prioritise *de novo* Fe-S cluster biogenesis to support haem synthesis and erythrocyte function as an adaptation to the vascular environment of the FGR placenta.

**Abstract** Fetal growth restriction (FGR) affects $\sim$10% of pregnancies worldwide and is often associated with placental insufficiency. Iron is essential for maternal haematopoietic adaptations and placental processes such as mitochondrial iron–sulphur (Fe-S) cluster assembly, haem synthesis and erythropoiesis. This study aimed to characterise iron transport and downstream utilisation in FGR. Placental tissues from term uncomplicated ($n = 19$) and FGR ($n = 18$) pregnancies were analysed. Maternal iron status was retrospectively assessed from clinical records. Placental mRNA and protein expression of iron-dependent pathways were analysed via RT-qPCR, LC-MS and western blotting. Placental iron content was assessed histologically, and haem levels were measured by an activity assay. FGR pregnancies showed significantly elevated maternal serum ferritin and lower red cell distribution width, although these remained within normal clinical values. Placental iron uptake transporters TFRC and DMT1 were significantly upregulated, while the iron exporter to the fetus, ferroportin, was reduced, indicating increased iron retention in the FGR placenta. Despite altered transporter expression, $Fe^{3+}$ iron levels were unchanged, suggesting iron utilisation over storage. Subsequent investigations identified reduced mitochondrial Fe-S synthesis components (FDXR, FDX2, NDUFAB1, HSPA9), and a prioritisation of mitochondrial and cytosolic haem synthesis enzymes in FGR. Protein levels of haemoglobin subunits (HBG1, HBG2, HBB, HBA1) and erythrocyte membrane markers (EPB41, EPB42, SPTA1, SPTB, ANK1) were decreased. These findings reveal a compensatory response in FGR placentae, with increased iron uptake and utilisation favouring haem synthesis over Fe-S cluster formation, possibly to support oxygen handling under poor placental vascularisation and reduced fetal oxygenation, with potential consequences for mitochondrial energy metabolism.

(Received 16 June 2025; accepted after revision 14 January 2026; first published online 17 February 2026)
**Corresponding author** J. J. Fisher: School of Medicine and Public Health, University of Newcastle, Callaghan, New South Wales, Australia. Email: Joshua.fisher@newcastle.edu.au

**Key points**

- Iron plays a critical role in placental function, and while iron-dependent pathway components are well-characterised, their integrated response and adaptive reprogramming in fetal growth restriction (FGR) remain poorly understood.
- In FGR, maternal iron status was unchanged, however, placental iron uptake proteins were increased and ferroportin reduced, suggesting that the placenta retains iron.
- FGR placentae showed altered *de novo* mitochondrial iron–sulphur cluster (Fe-S) formation and a bottleneck in late-stage Fe-S cluster assembly.
- This shift in Fe-S synthesis prioritises mitochondrial and cytosolic haem synthesis pathways, consistent with increased haem utilisation and breakdown.
- Globin subunits were lower in protein abundance and impaired placental erythrocyte structure in FGR.
- Dysregulation of erythrocyte membrane proteins in FGR placentae suggests altered erythrocyte structure, potentially representing an adaptive response to inadequate vascularisation, attempting to optimise oxygen delivery to the fetus.

## Introduction

Fetal growth restriction (FGR) occurs in 10% of pregnancies globally and is associated with increased perinatal and neonatal morbidity and mortality (Miller et al., 2008). FGR is defined by the inability of the fetus to achieve its genetically predetermined growth potential relative to its gestational age (Gordijn et al., 2016). Although maternal and fetal complications are recognised causes of FGR, 60–70% of FGR cases are associated with placental insufficiency, where the placenta fails to adequately deliver nutrients and oxygen. Placental insufficiency can arise from inadequate uterine spiral artery remodelling, impaired trophoblast invasion, abnormal vascular development, oxidative stress and inflammation (Brooker et al., 2025; Higashijima et al., 2013; Tang et al., 2017). In FGR placentae, the cellular landscape is altered, with impaired cytotrophoblast maturation and decreases in trophoblast volume. This results in a reduction in the number of placental villi and a decreased placental diameter (Sun et al., 2020). Consequently, the surface area available for oxygen and nutrient exchange is diminished, limiting substrate transfer to the fetus (Zygmunt et al., 2003). In response, the fetus adapts by redistributing blood flow, prioritising essential fetal organs, such as the brain and heart, resulting in asymmetrical growth and slowing the fetal growth rate (Miller et al., 2008; Roberts et al., 2020; Wu et al., 2023).

The placenta is a highly vascularised, potent haematopoietic organ. The placenta performs primitive haematopoiesis during embryogenesis, establishing a large reservoir of haematopoietic stem cells (HSCs) that not only support erythrocyte activity during early gestation, but also provide progenitor cells for the fetal liver and bone marrow when haematopoiesis transitions away from the placenta (Gekas et al., 2010; Palis & Segel, 1998). Haematopoietic function is critically reliant on iron availability, as iron is essential for erythropoiesis and mitochondrial functions that facilitate the biosynthesis of iron–sulphur clusters (Fe-S) and subsequent haem synthesis in erythroid cells (Chiabrando et al., 2014; Hentze et al., 2004; Mastrogiannaki et al., 2009).

Iron is an essential micronutrient for both fetoplacental development and mitochondrial bioenergetics (Dev & Babitt, 2017; Guo et al., 2019). During pregnancy, the requirement for iron alters as gestation advances. In early pregnancy, placental development occurs in a physiologically hypoxic environment, with relatively low iron demands (Guo et al., 2016). This low-oxygen environment stabilises hypoxia inducible factors (HIFs), which transcriptionally activate genes involved in oxygen and iron regulation, such as erythropoietin (EPO), hepcidin and erythroferrone (Semenza, 2014). In the first trimester, rising EPO levels support both maternal erythropoiesis and transient placental haematopoietic activity (Cindrova-Davies & Sferruzzi-Perri, 2022; Delaney et al., 2021; Fairchild Benyo & Conrad, 1999). Consequently, hepcidin is suppressed, enhancing maternal iron mobilisation and supporting the increase in maternal iron demands to meet increased cardiovascular, renal and haematological adaptations, as well as fetoplacental development (Zaugg et al., 2022). These adaptations require sufficient iron for enhanced haematopoiesis and haemoglobin synthesis. Subsequently, this supports optimal oxygen transfer, thereby sustaining fetal development, while also establishing fetal iron reserves in the later stages of gestation (Cao & Fleming, 2016; Roberts et al., 2020). Maternal iron deficiency, anaemia and excess iron alike have been implicated in adverse pregnancy outcomes, including FGR (Andrews, 2000), highlighting the important role of iron in healthy pregnancies.

Iron transport during pregnancy occurs unidirectionally from the mother to the fetus, primarily via transferrin-bound iron in the maternal circulation (McArdle et al., 2003). At the maternal placental interface, transferrin receptor 1 (TfR1/TFRC) binds the maternal transferrin–iron complex, inducing endocytosis (McArdle et al., 2011; Roberts et al., 2020). Within the endosome, the low pH induces a conformational change in TfR1, releasing ferric ($Fe^{3+}$) iron. Following the conversion of $Fe^{3+}$ to ferrous ($Fe^{2+}$) iron, iron is transported into the cytosol via divalent metal transporter 1 (DMT1) (McArdle et al., 2011; Roberts et al., 2020; Waldvogel-Abramowski et al., 2014). Once in the cytosol, iron has three potential fates, it is either: (i) stored in ferritin, (ii) shuttled to the mitochondria for Fe-S cluster biogenesis and haem synthesis, or (iii) transported across the basolateral (fetal) membrane via ferroportin for entry into the fetal circulation following

**Veronica Botha** is currently a PhD student working at the Hunter Medical Research Institute in Newcastle, Australia. Her research focuses on the role of iron in pregnancy, with a particular interest in how iron and oxygen sensing affect placental structure, mitochondrial function and fetal growth restriction. She is particularly interested in strengthening our understanding of fundamental placental biology.

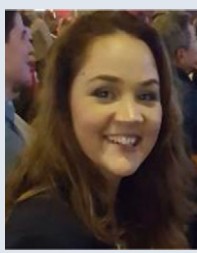

reoxidation to $Fe^{3+}$ (McArdle et al., 2011; Roberts et al., 2020). Approximately 20–50% of cellular iron is directed towards the mitochondria to support two critical pathways, the biosynthesis of Fe-S clusters and haem (Qi et al., 2023; Roberts et al., 2020)

Fe-S clusters are diverse molecular structures that perform various biological functions within the cell. Specifically, Fe-S clusters play a role within the mitochondrial electron transport chain (ETC), via redox reactions that transfer electrons and generate a proton gradient essential for ATP synthesis (Mostajabi Sarhangi & Matyushov, 2023; Paul et al., 2017). *De novo* Fe-S cluster synthesis begins with the formation of [2Fe-2S] clusters, which serve as essential precursors in the Fe-S cluster pathway. These *de novo* [2Fe-2S] clusters undergo further processing to generate more complex [3Fe-4S] and [4Fe-4S] clusters downstream. These late-stage [3Fe-4S] and [4Fe-4S] clusters are critical for the function and stability of the mitochondrial ETC, particularly in complexes I, II and III, where they facilitate efficient electron transfer during oxidative phosphorylation (Maio & Rouault, 2020; Mostajabi Sarhangi & Matyushov, 2023; Read et al., 2021). Additionally, the terminal enzyme in haem biosynthesis, ferrochelatase (FECH), requires [2Fe-2S] clusters for its activity, linking Fe-S cluster synthesis with haem production. Haem is essential for oxygen transport through its incorporation into haemoglobin and cytochromes, which facilitate electron transfer in the ETC, particularly complexes II, III and IV (Anderson & Frazer, 2017; Kim et al., 2012).

The primary utilisation of iron in fetal development is in erythropoiesis (Cerami, 2017), and haem stimulates the production of $\alpha$- and $\beta$-like globin genes with distinct haemoglobin profiles expressed at specific stages of gestation (Kim et al., 2012; Moksnes et al., 2022). Embryonic ($\zeta 2\varepsilon 2$) and fetal ($\alpha 2\gamma 2$) haemoglobin exhibit higher oxygen affinity compared to adult ($\alpha 2\beta 2$) haemoglobin, demonstrated by a leftward shift in their oxygen dissociation curves. These haemoglobin variants exhibit higher oxygen affinities during early development, enabling adaptation to lower oxygen availability and efficient maternal–fetal oxygen transfer (Manning et al., 2020).

Despite the well-described crucial role of iron in placental and fetal development, the specific alterations in iron-dependent pathways in FGR placentae remain uncharacterised. The complex interplay between iron transport, Fe-S cluster assembly and haem synthesis is particularly relevant in the context of FGR, where placental insufficiency and impaired oxygen delivery are defining clinical characteristics. We have therefore examined genes and proteins involved in placental iron transport, as well as iron-related mitochondrial proteins involved in Fe-S assembly, haem synthesis and placental erythropoiesis in FGR compared to healthy placentae. Collectively, these results provide molecular insights into how altered iron transport and iron-dependent pathways may contribute to mitochondrial dysfunction and placental insufficiency observed in FGR.

## Methodology

### Ethical approval

Ethical approval for this research was obtained from the University of Newcastle Human Research Ethics Committee (H-382-0602), Hunter New England Health Human Research Ethics Committee (02/06/12/3.13) and Site-Specific Assessment (SSA/15/HNE/291), in compliance with the *Declaration of Helsinki*. Informed written consent was obtained from all participants.

### Maternal and placental sample collection

Placental villous cores were collected from 19 healthy, uncomplicated and 18 FGR (percentiles: <3rd $n = 7$, <5th $n = 2$ and <10th $n = 9$) pregnancies within 45 min of birth as previously described by Fisher et al. (2024) and washed in ice-cold PBS to limit maternal blood contamination. Tissues were either preserved in formalin at 4°C overnight and stored in 0.1 M PBS with 0.05% sodium azide for histological analysis, or snap-frozen and stored at −80°C for later gene and protein analysis. FGR was diagnosed with the consensus criteria as published by Gordijn et al. (2016). All placentae collected for this study were healthy term placentae from singleton pregnancies delivered between 37 and 40 weeks of gestation either vaginally or by caesarean section. FGR placentae were collected from singleton pregnancies delivered by caesarean section at 34–40 weeks of gestation with an estimated fetal weight <10th centile and birthweight confirmed <10th centile. Exclusion criteria applied to all samples with premature membrane rupture, hypertension, diabetes, placental abnormalities, chorioamnionitis, immune deficiencies, infectious diseases, smoking, non-steroidal anti-inflammatory drug use, illicit drug use and metformin use were not eligible within our study. Maternal blood samples collected as part of routine care were analysed retrospectively to assess clinical iron parameters. Reference ranges were obtained using *New South Wales Health Pathology (NSWHP) Hospital Laboratory Clinical Projects Laboratory Manual* (NSW Health Pathology, 2019) in conjunction with *Pregnancy and Laboratory Studies: A Reference Table for Clinicians* (Abbassi-Ghanavati et al., 2009).

## Histological analysis of iron deposition by Prussian blue staining

To detect $Fe^{3+}$, healthy ($n = 7$; 4 males, 3 females) and FGR ($n = 7$; 3 males, 4 females, $n = 3 <3^{rd}$, $n = 1 <5th$, $n = 3 <10th$) placental tissue were embedded in paraffin and sectioned at a thickness of 4 µm using a Leica RM2135 Microtome (Leica Biosystems, Wetzlar, Germany). Fixed slides were dewaxed in a series of xylene (POCD Healthcare, North Ryde, NSW, Australia) and ethanol (POCD Healthcare, AUS) submersions and then washed in MilliQ water. Subsequently, samples were stained with 10% (w/w) Prussian blue (potassium hexacyanoferrate (II) trihydrate, Sigma-Aldrich, Castle Hill, NSW, Australia) for 5 min, then 5% (w/w) Prussian blue stain in 10% HCl (Sigma Aldrich, AUS) for 30 min, and counterstained with nuclear fast red for 5 min (Sigma Aldrich, AUS) as described by Zaugg et al. (2024). Slides were then washed and dehydrated following gradient ethanol exposure of 70% and 100% for 3 min, finishing with a 10 min xylene exposure. Slides were mounted with a coverslip and left to dry overnight. Slides were then imaged on an Aperio GT 450 DX (Leica Biosystems) at $40\times$ magnification. The intensity of the Prussian blue staining was quantified using H-scores obtained from Halo Quantitative Pathology analysis software (Indica Labs, Albuquerque, NM, USA). All images were processed using identical parameters to ensure valid comparisons between healthy and FGR placental samples.

## Placental RNA preparation and gene expression studies

Placental tissue was crushed on dry ice with liquid nitrogen and subsequently lysed and homogenised with 1 mL Trizol® Reagent (Life Technologies; Thermo Fisher Scientific, Brisbane, Australia) using the Precellys24 instrument (Bertin Technologies, Montigny-le-Bretonneux, France) for two cycles of 30 s at 5000 $g$, incubated on ice for 30 min and then repeated. Samples were centrifuged for 10 min at 10,000 $g$ using the Microfuge 20R (Beckman Coulter, Lane Cove, Australia). The remainder of the protocol was performed in line with the manufacturer's instructions outlined in the Direct-zol$^{TM}$ RNA mini-prep kit (Zymo Research, South San Francisco, CA, USA). Following quantification on the Nanodrop One (Thermo Fisher Scientific, AUS), 1000 ng of cDNA was synthesised using SuperScript$^{TM}$ IV First-Strand Synthesis System (Thermo Fisher Scientific, AUS). RT-qPCR was conducted using the PowerUpTM SYBR Green Master Mix (Applied Biosystems, Thermo Fisher Scientific, AUS) with KiCqstart primers (Sigma Aldrich, AUS; Table 1). RT-qPCR was performed using the QuantStudio6 Flex Real-Time PCR system (Thermo Fisher Scientific, Waltham, MA, USA). Relative gene expression was determined by normalising to the geometric mean of three housekeeping genes ($\beta$-actin, TBP and YWHAZ), which were quantified using the $2^{-\Delta\Delta CT}$ method to assess relative gene expression within placental tissue samples.

## Proteomics studies

Placental tissue was crushed on dry ice and stored at $-80°C$ until LC-MS-based proteomic analysis could be performed. LC-MS sample preparation followed previously published methodology by Mulhall et al. (2024), with placental tissue samples lysed in chilled 4% (w/v) sodium deoxycholate (SDC; Sigma Aldrich, AUS), 100 mM Tris-HCl (pH.8.5; Sigma Aldrich, AUS) and passed through an 18G blunt-end needle before sonification. Samples were quantified and concentrations equalised to 200 µg/µL before digestion, alkylation and overnight trypsin (Thermo Fisher Scientific, AUS) incubation. Following trypsin exposure, peptides were diluted 1:1 with 100 mM Tris-HCl (ensuring SDC concentration <1%) and vortexed briefly before adding 99% ethylacetate/1% trifluoroacetic acid (TFA; 1:1; Thermo Fisher Scientific, AUS) and thoroughly vortexed at 2000 rpm using ThermoMixer. Peptides were then desalted using styrene-divinylbenzene reverses-phase sulphonated StageTips.

Peptide separation was performed on an Aurora Ultimate C18 column (25 cm $\times$ 75 µm, IonOpticks, Abingdon, UK) using a 90 min linear gradient from 3% to 41% of solvent B (acetonitrile in 0.1% formic acid) at a flow rate of 400 nL/min. MS analysis was conducted on an Eclipse mass spectrometer (Thermo Fisher Scientific, AUS), and MS scans were acquired over a mass range of 375–1400 $m/z$ at a resolution of 60,000, with an automatic gain control (AGC) target of 100% and maximum injection time of 50 ms. MS/MS scans were acquired at a resolution of 15,000, with an AGC target of 100%, normalised collision energy of 30% and dynamic maximum injection time. Raw data were analysed using Proteome Discoverer 2.5 (Thermo Fisher Scientific, AUS), outlined by Murray et al. (2023).

Proteomic analysis employed a bottom-up approach to profile protein abundance (Murray et al. 2023). Log$_2$ fold change (FC) thresholds were not applied, to ensure that specific protein changes within relevant proteins of interest were not overlooked or masked by larger magnitude changes within the dataset, as previously published (Aebersold & Mann 2016; Bandeira et al. 2021; Geyer et al. 2016).

## Western blotting

To validate our proteomics data, western blotting was performed on the same subset of healthy ($n = 7$;

**Table 1. KiCqStart primers for gene expression studies of targeted iron transporters and intermediaries**

| Target gene name | Gene abbreviation | Primer sequence |
|---|---|---|
| Cellular iron transporters | | |
| Transferrin receptor 1 | TFRC | F 5′-AAGATTCAGGTCAAAGACAG-3′ |
| | | R 5′-CTTACTATACGCCACATAACC-3′ |
| Solute carrier family 11 member 2 | SLC11A2 | F5′-GAGTATGTTACAGTGAAACCC-3′ |
| | | R 5′-GACTTGACTAAGGCAGAATG-3′ |
| Solute carrier family 40 member 1 | SLC40A1 | F 5′-AAAGATACTGAGCCAAAACC-3′ |
| | | R 5′-GTTGTAGTAGGAGACCCATC-3′ |
| Mitochondrial iron transporters | | |
| Solute carrier family 25 member 37/mitoferrin 1 | SLC25A37 | F 5′-GGTAATGAATCCAGCAGAAG-3′ |
| | | R 5′-AGGAACTCATAGGTGATGAAG-3′ |
| Solute carrier family 25 member 28/mitoferrin 2 | SLC25A28 | F 5′-ACCTTTGATAACCTCTCTCC-3′ |
| | | R 5′-TATTAGGAACCAGGGGAATG-3′ |
| Mitochondrial haem synthesis | | |
| Coproporphyrinogen oxidase | CPOX | F 5′-CATGGAAATCTTTCAGAGGAAG-3′ |
| | | R 5′-ATTCTTGGGGTGGATAACAG-3′ |
| Ferrochelatase | FECH | F 5′-GGAAGAATATCCTCTTGGTTC-3′ |
| | | R 5′-CACTCCTTGGCTAAAACTTG-3′ |
| Erythropoiesis | | |
| Haemogen | HEMGN | F 5′-CAAGATGGATTTGGGAAAGG-3′ |
| | | R 5′-TCTGGAGAATGGTTCTCTTC-3′ |
| Haemoglobin subunit alpha | HBA | F 5′-GACCTCCAAATACCGTTAAG-3′ |
| | | R 5′-ACTTTATTCAAAGACCACGG-3′ |
| Haemoglobin subunit beta | HBB | F 5′-ATTTGCTTCTGACACAACTG-3′ |
| | | R 5′-CAAAGGACTCAAAGAACCTC-3′ |
| Haemoglobin subunit gamma 1 | HBG1 | F 5′-ACTTCCTTGGGAGATGCCAC-3′ |
| | | R 5′-GCCTATCCTTGAAAGCTCTGA-3′ |
| Haemoglobin subunit gamma 2 | HBG2 | F 5′-ACTTCCTTGGGAGATGCCAT-3′ |
| | | R 5′-GCCTATCCTTGAAAGCTCTGC-3′ |
| Erythrocyte membrane protein band 4.1 | EPB41 | F 5′-AAAGGACTTGGAAGGAGTAG-3′ |
| | | R 5′-GCGGTTAATTCTCAGCTTATC-3′ |
| Solute carrier family 4 member 1 | SLC4A1 | F 5′-CAAGAACAGCTCCTATTTCC-3′ |
| | | R 5′-CTGAATGAAGAAATCCACCAG-3′ |
| Iron–sulphur cluster biosynthesis | | |
| LYR motif containing 4 | LYRM4/ISD11 | F 5′-AGGATAAGAGATGCCTTCAG-3′ |
| | | R 5′-ATCTGTCGACGAATTACTCC-3′ |
| NADH: ubiquinone oxidoreductase subunit AB1 | NDUFAB1 | F 5′-CAGGTTCCTTGGTAGAGTTAC-3′ |
| | | R 5′-TGGGTCAATCTTGTCATAGAG-3′ |
| Ferredoxin 2 | FDX1L/FDX2 | F 5′-CATGTGTATGTGAGTGAAGAC-3′ |
| | | R 5′-TAGAAGGTTCCTGGTGATCTTG-3′ |
| Frataxin | FXN | F 5′- GAAAGACTAGCAGAGGAAAC-3′ |
| | | R 5′-TCCCAAAGGAGACATCATAG-3′ |
| Iron–sulphur cluster assembly enzyme | ISCU | F 5′- GTGTAGACCTTTCTGCTCAGG-3′ |
| | | R 5′-TAGATGTCTTGTCAAGGGAC-3′ |
| Glutaredoxin 5 | GLRX5 | F 5′-AAGAAGGACAAGGTGGTG-3′ |
| | | R 5′-TAGTCTTTAATGCCTTGTCG-3′ |
| NFU1 iron–sulphur cluster scaffold | NFU1 | F 5′-CCCTAGTTCAATCATTACTCTG -3′ |
| | | R 5′-CATCATCCATAACCTGTTCTAC- 3′ |
| NUBP iron–sulphur cluster assembly factor | NUBPL | F 5′-TGAAGTTCTAGGAGACATTCC-3′ |
| | | R 5′-TCTTCTTACCACTTCCACAG-3′ |
| BolA family member 3 | BolA3 | F 5′-CGAGCTACAGCTATAAAAGTC-3′ |
| | | R 5′-TTGGGGACAGAGGTAAATATC-3′ |
| Housekeeper genes | | |
| Beta actin | $\beta$-Actin | F 5′-GACGACATGGAGAAAATCTG-3′ |
| | | R 5′-ATGATCTGGGTCATCTTCTC-3′ |

*(Continued)*

**Table 1. (Continued)**

| Target gene name | Gene abbreviation | Primer sequence |
| --- | --- | --- |
| Tryptophan 5-monooxygenase activation protein zeta | YWHAZ | F 5′-CCTGCATGAAGTCTGTAACTGAG-3′<br>R 5′-GACCTACGGGCTCCTACAACA-3′ |
| TATA-box binding protein | TBP | F 5′-GCCAAGAGTGAAGAACAG-3′<br>R 5′-GAAGTCCAAGAACTTAGCTG-3′ |

KiCqStart primers targeting nuclear DNA involved in placental transport and iron-dependent pathways. Forward (F) and reverse (R) primers were reconstituted to a 100 µM final concentration following the manufacturer's instructions. Gene expression was analysed using the $2^{-\Delta\Delta CT}$ method. Genes were examined in response to the physiological function of iron in healthy and FGR placentae.

4 males, 3 females) and FGR ($n = 7$; 3 males, 4 females, $n = 3$ <3rd, $n = 1$ <5th, $n = 3$ <10th). Protein was extracted from placental tissue using 1 mL RIPA buffer [50 nM Tris-HCl (pH 7.4), 150 nM NaCl (Thermo Fisher Scientific, AUS), 1% NP-40 (Thermo Fisher Scientific, AUS), 1% sodium deoxycholate, 1% SDS, Complete Mini Protease Inhibitor Cocktail tablets (Roche Diagnostics, North Ryde, Australia) and 1 nM PMFS. Samples were homogenised using Precellys24 and centrifuged as outlined previously. The supernatant was collected and stored at −80°C for subsequent western blotting analysis. Following the protocol of Fisher et al. (2019), samples (40 µg/µL) were loaded onto 12% Tris gels (Invitrogen, Australia), electrophoresed and transferred to polyvinylidene fluoride (PVDF) membranes (Merck Millpore, Melbourne, VIC, Australia). Membranes were blocked with Odyssey Intercept buffer (LICORBio, Lincoln, NB, USA) for 1 h, then incubated overnight with primary antibody, LYMR4 at 1:250 dilution (ab253001, Abcam, Australia) and Beta Actin (ab8226, Abcam) at a 1:1000 dilution. After washing with PBST, membranes were incubated for 1 h with secondary antibodies anti-mouse IRDye®800CW and anti-rabbit IRDye®680LT (LICORBio, USA) at 1:20,000 dilution and washed with PBST. Membranes were imaged using the LI-COR Odyssey M system (LICORBio, USA) and analysed with Image Studio v5.2.

### Assessment of haem

Placental haem concentrations were measured using a haem assay kit (MAK316, Sigma Aldrich). Following the manufacturing instructions, a total of 250 µL of water was added to a 96-well plate, creating a blank well. A standard well was created by mixing 50 µL of haem calibrator and 200 µL of water. Subsequently, 50 µL of homogenised term and FGR samples suspended in glycerol buffer (50 mM Tris-HCl, 1 mM EDTA, 0.5% Glycerol) was mixed with 200 µL of reagent respectively. The plate was left to incubate for 5 min at room temperature, and the absorbance of well contents was determined at 400 nm. All samples and standards were performed in duplicate,

and total haem concentration was determined using the following equation:

$$[\text{Total Haem}]\left(\frac{\mu\text{mol}}{\text{L}}\right) = \frac{(\text{ODsample} - \text{ODblank})}{(\text{ODcalibrator} - \text{ODblank})} \times 62.5\left(\frac{\mu\text{mol}}{\text{L}}\right)$$

### Statistical analysis

All data were analysed using GraphPad Prism 10.0. Analysis of fetal sex was not conducted due to insufficient sample size, but for transparency, female data points are represented by open shapes in the figures, while closed shapes represent male data points. Outliers were identified using the ROUT test ($Q = 1\%$), which detects multiple outliers while controlling the false discovery rate (Motulsky & Brown, 2006). Data normality was assessed using the Shapiro–Wilk test. For normally distributed data, Student's $t$ tests were performed. For data that did not follow a normal distribution, Welch's $t$ tests were conducted. Statistical significance was set at $P < 0.05$.

John Hunter Hospital clinical data are presented with 95% confidence intervals (CIs) and interquartile ranges (IQRs). Gene expression data are presented as mean ± standard deviation (SD), and histological H-scores, accounting for blue-stained iron deposits within placental tissue, were quantified using Halo Quantitative Pathology Analysis Software (Indica Labs) and calculated using the following formula (Wen et al., 2024):

$$H - \text{score} = (3 \times \% \text{ heavily stained}) + (2 \times \% \text{ moderately stained}) + (1 \times \% \text{ lightly stained})$$

This formula weights the staining intensity, generating a composite score ranging from 0 to 300, reflecting both the extent and intensity of iron deposits. Proteomic analysis was expressed $\log_2$ fold change (FC) to quantify differential protein abundance between healthy controls and FGR, visualised using heatmaps. The $\log_2$ FC was

derived from the ratio between healthy term placentae to assess changes in FGR placentae, as such a positive $\log_2$ FC indicates lower protein abundance in FGR placentae, while a negative $\log_2$ FC indicates higher protein abundance in FGR placentae. Proteins with fewer than two peptide spectrum matches (PSMs) were excluded from the analysis to ensure data reliability.

## Results

### Clinical insights into maternal iron bioavailability, birth and gestational characteristics

To better understand the potential role of iron throughout gestation, we evaluated clinical characteristics encompassing maternal physical attributes, birth-related factors and gestational characteristics (Table 2).

**Table 2. Maternal, fetal and birth clinical characteristics from pregnancies diagnosed as FGR and healthy term controls**

| | Term control | FGR | *P* value |
|---|---|---|---|
| *N* | 19 | 18 | |
| (a) Maternal and birth characteristics | | | |
| Maternal age (years) | 31 (27.8, 33.0) | 28.5 (26.9, 31.0) | 0.3631 |
| Maternal body mass index; BMI (kg/m$^2$) | 26.0 (24.2, 28.2) | 25.0 (22.4, 30.6) | 0.8836 |
| Birth weight (g) | 3555 (3346, 3676) | 2415 (1760, 2356) | **<0.0001** |
| Placental weight (g) | 689 (628, 757) | 556 (395, 519) | **<0.0001** |
| Sex ratio | 9 Males, 10 Females | 8 Males, 10 Females | — |
| (b) Fetal biometry measurements following sonography assessment | | | |
| Fetal abdominal circumference (mm) | 330 (321, 339) | 275 (257, 293) | **<0.0001** |
| Fetal femur length (mm) | 68.4 (66.3, 73.6) | 61.6 (58.0, 65.2) | **0.002** |
| Fetal head circumference (mm) | 328.0 (315.2, 347.6) | 295.5 (282.4, 307.8) | **0.0007** |
| Amniotic fluid volume (cm) | 12.6 (10.8, 15.4) | 10.9 (7.99, 12.1) | 0.054 |
| UAD pulsatility index | 0.950 (0.820,1.50) | 1.10 (0.860, 1.41) | 0.823 |
| UAD resistance index | 0.585 (0.530, 0.790) | 0.620 (0.560, 0.730) | 0.5476 |
| (c) Maternal haematological parameters | | | |
| Haemoglobin (Hb) RI: 10.5–15.0 g/dL | 12.7 (12.1, 13.0) | 12.2 (11.8,12.7) | 0.325 |
| Mean corpuscular haemoglobin (mcH) RI: 29.0–33.0 pg per cell | 30.0 (28.4, 30.4) | 30.0 (28.9, 30.8) | 0.519 |
| Mean corpuscular haemoglobin concentration (mcHc) RI:310–360 g/L | 345 (338, 352) | 344 (338, 346) | 0.437 |
| Haematocrit (HCt) RI: 38.0–41.0% | 37.6 (36.1,39.5) | 37.1 (35.9, 38.7) | 0.620 |
| Mean corpuscular volume (McV) RI: 81–99 μm$^3$ | 88.0 (86.1, 90.0) | 88.0 (86.4, 90.4) | 0.8 |
| Red cell count (RCC) RI: 2.81–4.55 10$^6$/mm$^3$ | 4.09 (4.02, 4.34) | 4.00 (3.86, 4.13) | 0.081 |
| Red cell distribution width (RDW) RI: 12.5–15.3% | 14.2 (13.8, 15.0) | 13.4 (13.1, 13.7) | **0.006** |
| (d) Maternal haematocrit and anaemia screening | | | |
| Iron (Fe) RI: 5.37–34.55 μmol/L | 22.0 (14.5, 24.4) | 15.50 (12.1, 20.4) | 0.29 |
| Transferrin (Tf) RI: 2.2–5.3 g/dL | 3.50 (2.97, 4.12) | 3.20 (2.82, 3.76) | 0.513 |
| Ferritin (F) RI: 30–200 μg/L | 20.0 (14.8, 33.3) | 46.0 (25.4, 67.7) | **0.029** |

This table presents data from consenting participants at John Hunter Hospital. Data are presented as median and interquartile range (IQR). Maternal haematological reference intervals (RI) were obtained from NSW Health Pathology (2019) and Abbassi-Ghanavati et al. (2009). Shapiro–Wilk normality tests were conducted, with Student's *t* tests or Welch's tests used where appropriate. Statistically significant values ($P < 0.05$). Missing data are indicated by —.

Analysis included maternal parameters (age and body mass index (BMI)), birth outcomes (birth weight, birth centile, placental weight, fetal sex) and fetal biometry measurements following sonography assessment (estimated fetal weight (EFW%), abdominal circumference, femur length, head circumference and amniotic fluid volume). FGR babies were significantly smaller ($P < 0.0001$), with a median birth weight of 2415 g (95% CI: 1760—2356 g), compared to healthy term babies, who had a median birth weight of 3555 g (95% CI: 3346–3676 g). Additionally, FGR babies had smaller placentae ($P < 0.0001$), with a median placental weight of 556 g (95% CI: 395—519 g), compared to healthy term babies, who had a median placental weight of 689 g (95% CI: 628—757 g). Furthermore, fetal biometric measurements, including abdominal circumference ($P < 0.0001$), femur length ($P = 0.002$) and head circumference ($P = 0.0007$), were all significantly lower in FGR pregnancies. Although the difference in amniotic fluid volume was not statistically significant, FGR pregnancies had lower amniotic fluid volume ($P = 0.054$) compared to healthy term controls.

Maternal haematological indices and maternal haematocrit and anaemia screening (Table 2) were performed on healthy and FGR pregnancies. No clinical or statistical significance was observed between FGR and healthy control groups in haemoglobin (Hb), mean corpuscular haemoglobin (MCH), mean corpuscular haemoglobin concentration (MCHC), mean corpuscular volume (MCV), red cell count (RCC), haematocrit (HCt), systemic iron levels and transferrin (Tf). Maternal ferritin levels were higher ($P = 0.029$) and red cell distribution width (RDW) lower ($P = 0.006$) in FGR cases compared to healthy controls, but both parameters fell within clinical reference ranges.

### Placental iron transport

Analysis of iron transporters between FGR and control placentae revealed significant alterations in gene expression and protein levels of iron import and export components. TFRC showed significantly increased mRNA expression ($P = 0.0063$, Fig. 1*A*) and upregulated protein levels ($-0.117$ log$_2$ FC; Fig. 1*D*) in FGR placentae compared to healthy term placentae. DMT1/SLC11A2 exhibited significantly increased mRNA expression in FGR ($P = 0.0110$, Fig. 1*B*) compared with controls. In contrast, ferroportin (FPN/SLC40A1; Fig. 1*C*), the iron receptor responsible for transporting iron out of the placenta into fetal circulation, showed no significant difference ($P = 0.0527$) in mRNA expression between FGR and healthy controls. FPN protein analysis (Fig. 1*D*) revealed lower protein levels (0.641 log$_2$ FC) in FGR placental villous tissue. These findings suggest FGR placentae are retaining iron for placental function.

Following the clinical characterisation of maternal iron parameters and mRNA expression of iron transport into the placenta, the placental iron content between healthy and FGR placentae was assessed. Histological analysis by Prussian blue staining, using the same subset of control ($n = 7$; 4 males, 3 females) and FGR ($n = 7$; 3 males, 4 females, $n = 3$ <3rd, $n = 1$ <5th, $n = 3$ <10th) placentae analysed for western blotting and proteomics, revealed intense blue regions with no statistical difference ($P = 0.262$) between healthy control (Fig. 1*F–H*) and FGR (Fig. 1*I–K*) placentae.

### Mitochondrial iron transport

Given the pivotal role of iron in mitochondrial function, we examined mitochondrial iron transporters located within the inner mitochondrial membrane, mitoferrin 1 (solute carrier family 25 member 37, SLC25A37) and mitoferrin 2 (solute carrier family 25 member 28, SLC25A28). No significant difference in mitoferrin 1 ($P = 0.2396$; Fig. 2*A*) mRNA expression was observed in FGR placentae compared to healthy controls. In contrast, mitoferrin 2 (Fig. 2*B*; $P = 0.0012$) mRNA expression was significantly lower in FGR placental villous tissue than in healthy controls.

### Mitochondrial iron utilisation: *de novo* Fe-S cluster synthesis

The utilisation of iron within the mitochondria converges on the synthesis of Fe-S clusters. This process comprises two main stages: *de novo* synthesis of [2Fe-2S] clusters and late-stage genesis of [4Fe-4S] clusters vital to mitochondrial function and cellular viability. In *de novo* Fe-S cluster synthesis, the mRNA expression of LYR motif 4 (LYRM4/ISD11; Fig. 2*C*), NADH: ubiquinone oxidoreductase subunit AB1 (NDUFAB1/ACP; Fig. 2*D*) and frataxin (FXN; Fig. 2*E*) were similar in FGR and healthy control placental villous tissue. Conversely, protein levels of NDUFAB1 (0.123 log$_2$ FC; Fig. 2*L*) were lower in FGR placentae compared with healthy controls. Ferredoxin 2 (FDX2/FDX1L) mRNA expression (Fig. 2*G*) was significantly lower ($P = 0.0138$), while ferredoxin reductase (FDXR; Fig. 2*L*) was 1.08 log$_2$ FC lower in protein levels in FGR compared to controls. The mRNA expression of iron–sulphur cluster assembly scaffold enzyme (ISCU; Fig. 2*F*), which is essential for [2Fe-2S] assembly and transfer, remained unchanged between FGR placental tissue compared with healthy controls. However, ISCU protein expression ($-0.372$ log$_2$ FC; Fig. 2*L*) was higher in FGR placental villous tissue when compared with healthy controls. An important transporter of the [2Fe-2S] cluster, glutaredoxin 5 (GLRX5; Fig. 2*H*), had significantly lower mRNA expression ($P = 0.0065$) in FGR

placentae; however, no difference in protein abundance between FGR and healthy term placentae was observed.

## Late-stage Fe-S cluster synthesis

In late-stage Fe-S cluster assembly, heat shock protein family A (HSP70) member 9 (HSPA9; Fig. 2L) showed a 0.219 log$_2$ FC increase at the protein level in healthy controls compared to FGR placentae. IBA57 (Fig. 2L) was $-0.073$ log$_2$ FC higher at the protein level, with statistical significance ($P = 0.001$) confirmed by western blotting (Fig. 2M). Increased mRNA expression was observed in [4Fe-4S] cluster transporters NFU1 iron–sulphur cluster

scaffold (NFU1; $P = 0.0120$; Fig. 2I), BOLA Family member 3 (BolA3; $P = 0.0148$; Fig. 2J), while NUBP iron–sulphur cluster assembly factor (NUBPL; $P = 0.0118$; Fig. 2K) mRNA expression was decreased.

## Haem synthesis

Analysis of the haem synthesis pathway showed significantly lower mRNA expression of coproporphyrinogen oxidase (CPOX; $P = 0.002$; Fig. 3A) in FGR placental villous tissue, yet protein levels were higher ($-0.163$ log$_2$ FC; Fig. 3D). In contrast, mRNA expression of ferrochelatase (FECH; Fig. 3B) at the gene

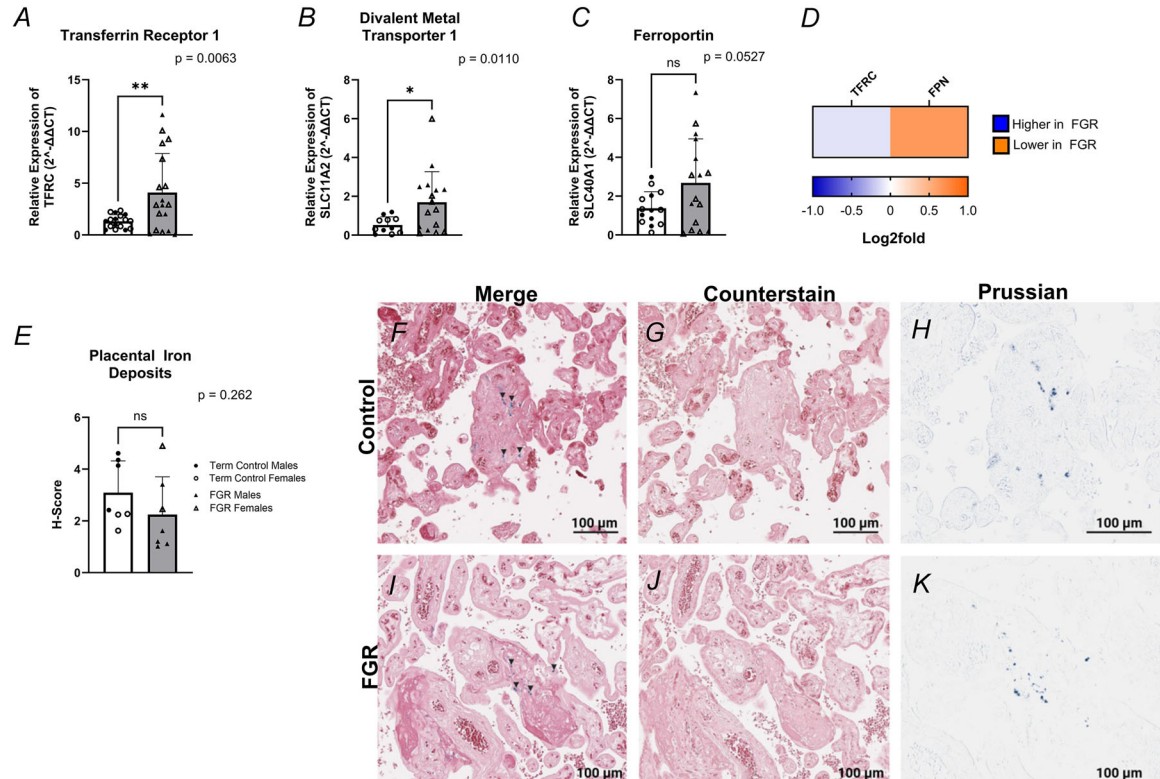

**Figure 1. mRNA expression and protein levels of cellular iron transporters and histological Prussian blue staining in FGR and healthy control placental villous tissue**
Female data points are represented by open shapes, while closed data points represent males. *A*, TFRC, transferrin receptor 1; *B*, SLC11A2, solute carrier family 11 member 2/divalent metal transporter 1 (DMT1); *C*, SLC40A1, solute carrier family 40 member 1/ferroportin (FPN). Data are presented as mean ± SD, control $n = 11$–19 (white and circles), FGR $n = 10$–18 (grey and triangles). Student's $t$ tests were performed: *$P < 0.05$, **$P < 0.01$, ***$P < 0.001$, ****$P < 0.0001$. *D*, proteomes of cellular iron transport depicted as a heatmap based on log$_2$ FC, with transporters indicated on top. The log$_2$ FC was derived from the ratio between healthy term placentae to assess changes in FGR placentae, so a negative log$_2$ FC indicates higher protein abundance in FGR placentae (blue), and a positive log$_2$ FC indicates lower protein abundance in FGR placentae (orange). *E*, Halo Pathology Analysis software was used to quantify the percentage of Prussian blue staining relative to the tissue area. Heavy, moderate and light intensity values were used to calculate the H-score in healthy control ($n = 7$) and FGR ($n = 7$) placental tissue. An unpaired $t$ test revealed no significant difference between the groups. *F*, a representative merged image of healthy control placental tissue showing Fe$^{3+}$ deposits (blue staining indicated by arrows). *G*, control placental tissue stained with nuclear fast red counterstain only; and *H*, Prussian blue staining isolated from counterstained tissue in *F*, upon which intensity calculations were performed. *I*, a representative merged image of FGR placental tissue, with *J* and *K* shown similarly to the row above (*G* and *H*). Scale bar = 100 μm.

($P = 0.0036$) and protein level ($-0.356$ log$_2$ FC; Fig. 3D) were higher in FGR relative to controls. Furthermore, protein levels of additional enzymes important to haem synthesis, uroporphyrinogen decarboxylase (UROD; Fig. 3D) and hydroxymethylbilane synthase (HMBS; Fig. 3D), were $-0.175$ log$_2$ FC and $-0.196$ log$_2$ FC greater in FGR, respectively. In contrast, aminolevulinate dehydratase (ALAD/PBGS; Fig. 3D) showed 0.541 log$_2$ FC lower protein expression in FGR compared with

term control placental villous tissue. Although haem synthesis enzymes were upregulated, this was not conserved when assessing haem activity ($P = 0.001$) in FGR placentae. Haem oxygenase (HMOX) -1 protein abundance was $-1.13$ log$_2$ FC higher in FGR placentae compared to control placentae (Fig. 3D). Additionally, NADPH-cytochrome P450 oxidoreductase (POR) protein abundance was $-0.157$ log$_2$ FC higher in FGR placentae than healthy controls.

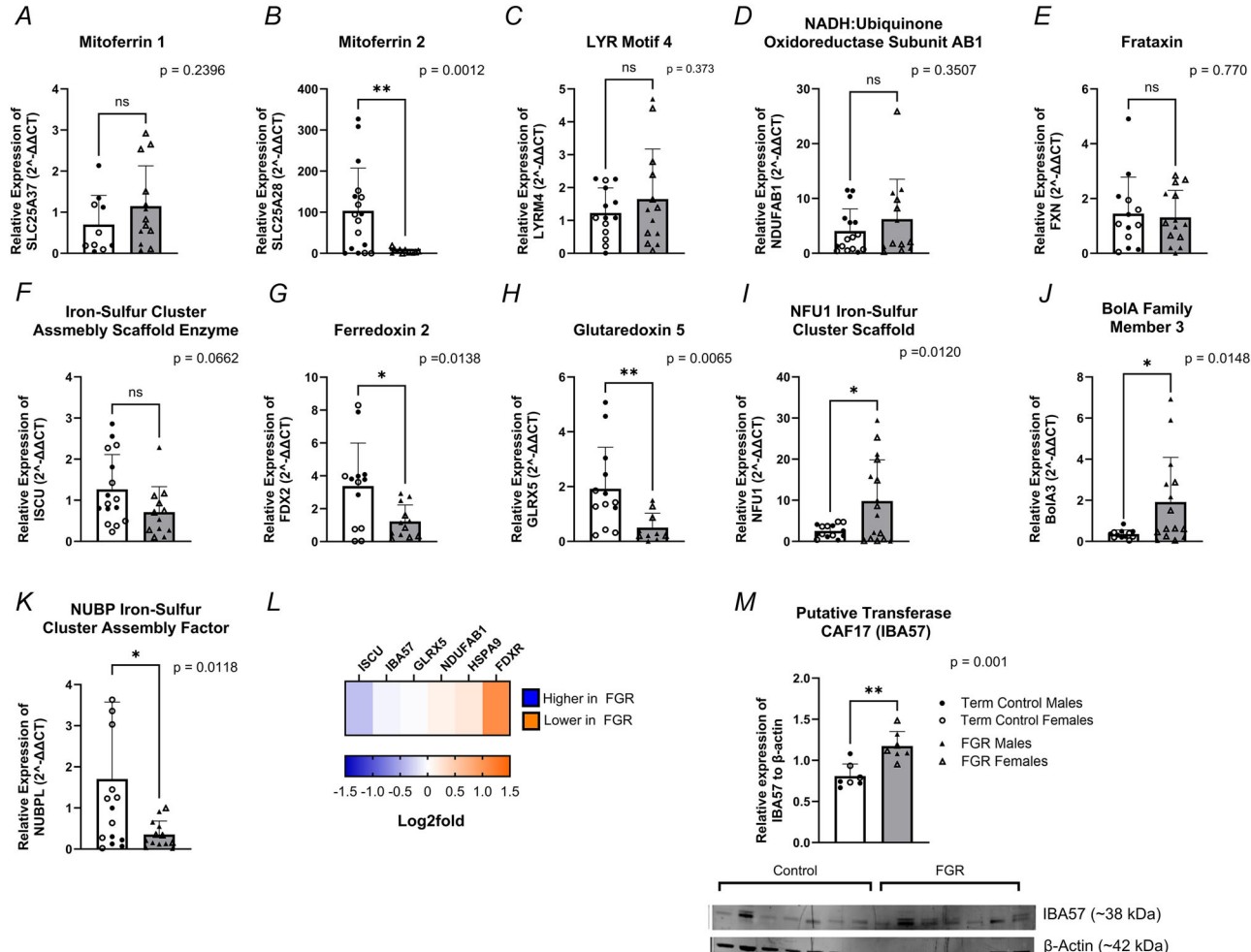

**Figure 2. mRNA expression and proteomes of targeted genes in the iron–sulphur (Fe-S) cluster biosynthesis pathway**

Female data points are represented by open shapes, while closed data points represent males. *A*, SLC25A37, solute carrier family 25 member 37/mitoferrin 1; *B*, SLC25A28, solute carrier family 25 member 28/mitoferrin 2; *C*, LYRM4, LYR motif containing 4 (ISD11); *D*, NDUFAB1, NADH: ubiquinone oxidoreductase subunit AB1; *E*, ISCU, iron–sulphur cluster assembly enzyme; *F*, FXN, frataxin; *G*, FDX1l, ferredoxin 2 (FDX2); *H*, GLRX5, glutaredoxin 5; *I*, NUBPL, NUBP iron–sulphur cluster assembly factor; *J*, NFU1, NFU1 iron–sulphur cluster scaffold; *K*, BolA3, BolA family member 3. Data are presented as mean ± SD, control $n = 11$–$19$ (white and circles), FGR $n = 10$–$18$ (grey and triangles). Student's *t* tests were performed: *$P < 0.05$, **$P < 0.01$, ***$P < 0.001$, ****$P < 0.0001$. *L*, Fe-S cluster proteomes, depicted as a heatmap based on log$_2$ FC with proteome identification on top. Proteomes of Fe-S cluster assembly are depicted as a heatmap based on the log$_2$ FC, with transporters indicated on top. The log$_2$ FC was derived from the ratio between healthy term placenta to assess changes in FGR placentae, so a negative log$_2$ FC indicates higher protein abundance in FGR placentae (blue), and a positive log$_2$ FC indicates lower protein abundance in FGR placentae (orange). *M*, protein abundance of putative transferase CAF17 (IBA57; 38.155 kDa) normalised to $\beta$-actin (41.737 kDa) in healthy and FGR placentae. A representative immunoblot is shown below.

The placenta is a potent haemopoietic site where haem synthesis stimulates globin production, forming a haemoglobin tetramer of alpha- and beta-like chains critical for binding and transporting oxygen. Our analysis revealed no changes in the mRNA expression of haemoglobin subunit alpha 1 (HBA1; $P = 0.1801$) between controls and FGR placental villous tissue, although protein levels were 0.210 log$_2$ FC lower in FGR compared to healthy controls (Fig. 4*A* and *G*). Within the beta-like globin chains, we observed significantly greater mRNA expression of haemoglobin subunit gamma 1 (HBG1; $P = 0.0368$, Fig. 4*B*) in FGR placentae, although haemoglobin subunit gamma 2 (HBG2) showed no differences ($P = 0.616$) between control and FGR placentae. Protein abundance of HBG1 (0.325 log$_2$ FC) and HBG2 (0.092 log$_2$ FC) was lower in FGR compared with term control placental tissue samples (Fig. 4*G*). Haemoglobin subunit beta (HBB) mRNA expression was significantly higher ($P = 0.0397$) in FGR compared to healthy controls (Fig. 4*D*). However, HBB protein abundance was lower (0.585 log$_2$ FC) within FGR placental tissue when compared to healthy placental tissue samples (Fig. 4*G*).

Levels of erythrocyte structural proteins, specifically erythrocyte protein band 4.1 (EPB41), EPB42, spectrin alpha 1 (SPTA1), spectrin beta (SPTB), ankyrin1 (ANK1) and solute carrier family 4 member 1 (SLC4A1), were subsequently assessed. The mRNA expression ($P = 0.0144$) and protein abundance (0.254 log$_2$ FC) of EPB41 were lower in FGR placental tissue compared to healthy tissue controls (Fig. 4*E* and *G*). EPB42 protein abundance was 0.562 log$_2$ FC lower within FGR placentae than healthy placentae (Fig. 4*G*). Analysis of spectrin protein levels, SPTA1 and SPTB (Fig. 4*G*), was 0.156 log$_2$ FC and 0.491 log$_2$ FC lower in FGR placenta compared with healthy term placenta. Similarly, ANK1 and SLC4A1 protein abundance were 0.661 log$_2$ FC and 0.493 log$_2$ FC lower in FGR placentae than healthy placentae, suggesting compromised erythrocyte structure. However, analysis of haemogen (HEMGN; $P = 0.0969$; Fig. 4*F*), a regulator of haematopoietic cell differentiation, mRNA expression was similar in healthy and FGR placental villous tissue, indicating that these changes were not occurring at the onset of haematopoiesis but rather at erythrocyte maturation.

## Discussion

This project sought to elucidate the complex role of iron in placental and mitochondrial function in FGR. We identified altered placental iron transporter levels alongside disrupted mitochondrial Fe-S cluster

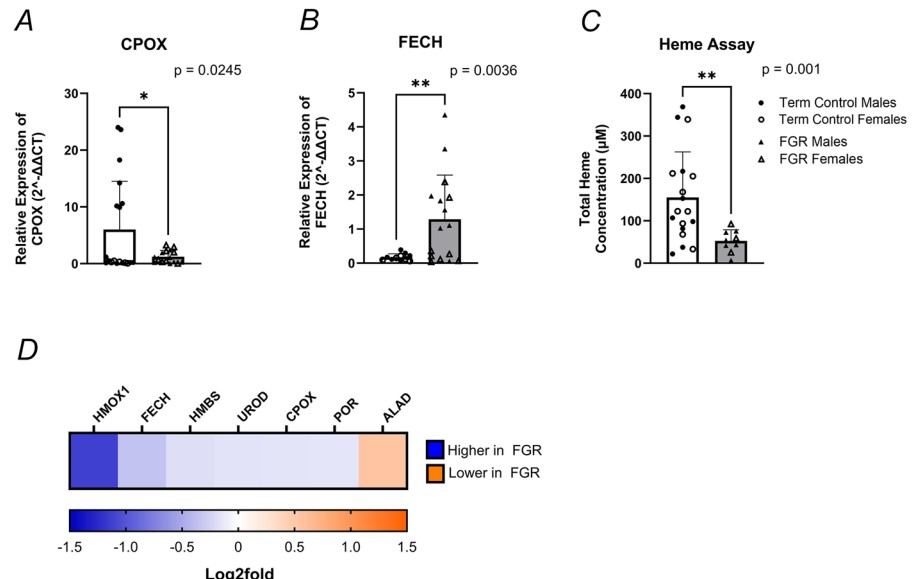

**Figure 3. mRNA expression of targeted genes involved in the haem synthesis pathway and haem content in FGR and healthy term placentae**

Female data points are represented by open shapes, while closed data points represent males. *A*, CPOX, coproporphyrinogen oxidase; *B*, FECH, ferrochelatase. Data are presented as mean ± SD, control $n = 11$–19 (white and circles), FGR $n = 16$–18 (grey and triangles), with sample loss due to outlier tests conducted. Student's *t* tests were performed: *$P < 0.05$, **$P < 0.01$, ***$P < 0.001$, ****$P < 0.0001$. *C*, haem assay measuring total haem concentration (μM) present. *D*, haem synthesis proteomes depicted as a heatmap based on log$_2$ FC, with proteome identification on top. Proteomes of haem synthesis depicted as a heatmap based on log$_2$ FC, with transporters indicated on top. The log$_2$ FC was derived from the ratio between healthy term placentae to assess changes in FGR placentae, so a negative log$_2$ FC indicates higher protein abundance in FGR placentae (blue), and a positive log$_2$ FC indicates lower protein abundance in FGR placentae (orange).

biosynthesis pathways in FGR placentae, specifically reduced capacity for [4Fe-4S] cluster synthesis due to lower levels of FDXR and FDX2. This suggests a metabolic redirection from incorporation into the mitochondrial ETC towards haem production, as evidenced by increased haem synthesis proteins. However, despite this apparent shift, total haem concentrations in FGR placentae were decreased. We propose that this arises from increased regulation of the haem degradation pathway in FGR placentae, evidenced by elevated protein levels of haem oxygenases and NADPH-cytochrome P450 oxidoreductase. Consequently, we observed reduced haemoglobin proteins within the FGR placentae. Additionally, our findings identified alterations in levels of erythrocyte structural proteins in FGR placentae. These alterations may further compromise oxygen delivery to the fetus or simultaneously represent an adaptive response to the poor vasculature of FGR placentae. Together, these findings enhance our understanding of the fundamental biological processes underlying FGR and highlight how adaptations in iron utilisation contribute to mitochondrial dysfunction observed in FGR, perpetuating inadequate fetal growth.

## Maternal haematological assessment in FGR

In FGR pregnancies, maternal haemodynamic responses are known to be compromised, including inadequate blood volume expansion and an impaired reduction in vascular resistance (Mecacci et al., 2021; Perry et al., 2020). Therefore, our study assessed whether maternal haemodynamic responses in FGR pregnancies might be reflected in maternal blood parameters. We observed reduced maternal RDW in FGR mothers compared with controls. RDW is the measurement of variation in red blood cell size and volume relative to the MCV (Fava et al., 2019). Despite being statistically significant, the reduction in maternal RDW remained within clinical reference ranges, indicating decreased anisocytosis, less variability in erythrocyte size and a narrower distribution around the MCV (Lippi & Plebani, 2014). However, RDW should be interpreted cautiously in the presence of inflammation (Owoicho et al., 2022). We speculate that the difference in maternal RDW between FGR and healthy control mothers, while within clinical reference ranges, may be a maternal adaptation to overcome poor spiral artery remodelling. Improved erythrocyte uniformity may enhance blood rheology (Nader et al., 2019; Zhang et al., 2009), which may provide advantages in the context

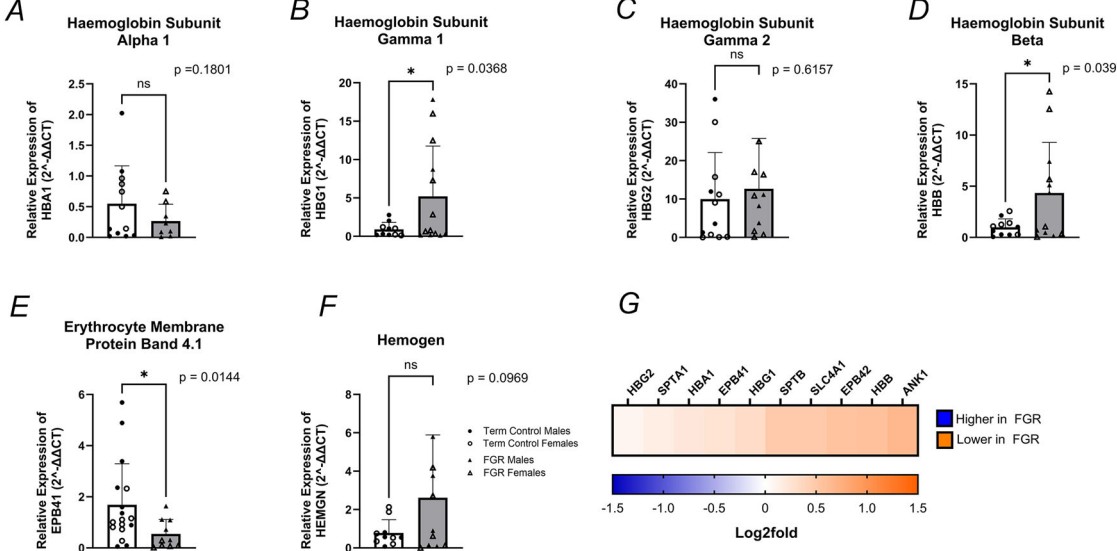

**Figure 4. mRNA expression and proteome of targeted genes involved in haematopoiesis**
Female data points are represented by open shapes, while closed data points represent males. *A*, HBA1, haemoglobin subunit alpha 1; *B*, HBG1, haemoglobin subunit gamma 1; *C*, HBG2, haemoglobin subunit gamma 2; *D*, HBB, haemoglobin subunit beta; *E*, EPB41, erythrocyte membrane protein band 4.1; *F*, HEMGN, haemogen. Data are presented as mean ± SD, control *n* = 10–19 (white and circles), FGR *n* = 8–18 (grey and triangles). Student's *t* tests were performed: *$P < 0.05$, **$P < 0.01$, ***$P < 0.001$, ****$P < 0.0001$. *G*, haematopoiesis proteomes, depicted as a heatmap and based on log$_2$ FC with proteome identification on top. Proteomes of haematopoiesis depicted as a heatmap based on log$_2$ FC, with transporters indicated on top. The log$_2$ FC was derived from the ratio between healthy term placentae to assess changes in FGR placentae, so a negative log$_2$ FC indicates higher protein abundance in FGR placentae (blue), and a positive log$_2$ FC indicates lower protein abundance in FGR placentae (orange).

of increased vascular resistance and narrowed maternal spiral arteries (Burton & Jauniaux, 2018).

Maternal serum ferritin levels were significantly higher in FGR pregnancies than healthy controls, although ferritin levels fell within clinical reference ranges (30–200 μg/L). Our observations are substantiated by previous studies showing that serum ferritin levels are increased in FGR pregnancies (Hou et al., 2000; Ozgu-Erdinc et al., 2014). The increase in maternal serum ferritin may therefore be attributed to an inflammatory response as ferritin acts as an acute-phase reactant during inflammation (Camaschella, 2015), and pregnancy is inherently pro-inflammatory.

Despite statistically elevated maternal serum ferritin levels and reduced RDW in FGR, the absence of maternal changes in iron-related haematological parameters outside clinical reference ranges suggests that subsequent findings were not due to impaired iron delivery. Instead, they may represent maternal adaptations as secondary responses to compromised placental function.

### Iron transport in the FGR placenta

Our study revealed that iron transporter proteins were altered in FGR placentae, characterised by increased iron uptake capacity. Specifically, we observed an increase in transferrin receptor 1 (TfR1/TFRC) levels, the major importer of non-haem iron into cells, in FGR placentae. While maternal iron deficiency has been shown to increase TfR1 expression in placental tissue (Sangkhae et al., 2020), ensuring appropriate iron uptake from the maternal circulation, we found no evidence of maternal iron deficiency within our FGR group. However, Burton and Fowden (2012) and Sibley et al. (2010) report that when placental size and capacity are reduced, the placenta increases nutrient transporter activity as a compensatory mechanism to suboptimal conditions. This may explain our observed increases in iron transporters in significantly smaller FGR placentae. Collectively, this suggests that the increase in TfR1 in our FGR placentae might represent a compensatory response to maximise iron uptake in a compromised placental vascular environment.

As iron transport is transcellular and unidirectional from mother to fetus through the placenta, the fetus depends entirely on maternal iron transfer mediated by ferroportin (FPN), which exports iron into the fetal circulation (Sangkhae et al., 2020). We observed lower FPN levels in FGR placentae, suggesting that the placenta is retaining iron for its own use (O'Brien, 2022; Sangkhae et al., 2020). This is supported by an increase in divalent metal transporter 1 (DMT1/SLC11A2) mRNA expression, the transporter of iron into the cell cytoplasm following endocytosis of the TfR1–iron complex. Sub-sequent Prussian blue staining showed no difference in iron deposit content for ferric ($Fe^{3+}$) iron between FGR and healthy controls. Collectively, these findings suggest that iron is being prioritised for intracellular function in FGR placentae, rather than for storage or export into the fetal circulation.

### Mitochondrial iron transport

The mitochondria utilise between ∼20 and 50% of cellular iron (Qi et al., 2023), facilitating mitochondrial function and bioenergetic pathways to support fetoplacental growth (Aye et al., 2022; Paul et al., 2017). Mitoferrin-1 (SLC25A37) and mitoferrin-2 (SLC25A28) facilitate iron transport across the inner mitochondrial membrane but exhibit different cell expression profiles and cell-specific functions (Ali et al., 2022). We observed no changes in mitoferrin-1 mRNA expression between our groups and propose that the expression of mitoferrin-1 is likely to be conserved, attributed to the placenta's fundamental role as a haematopoietic organ, rather than solely as a facilitator of maternal–fetal exchange. This is further demonstrated by Shaw et al. (2006), who observed that knocking out mitoferrin-1 in mice results in embryonic lethality at the initiation of definitive erythrocyte development. In contrast, Shaw et al. (2006) and Paradkar et al. (2009) found that mitoferrin-2 is ubiquitously expressed in non-erythroid tissue and is responsible for the iron import required for haemoproteins and Fe-S cluster synthesis (Ali et al., 2022). Our study observed a decrease in mitoferrin-2 mRNA expression in placentae from FGR pregnancies. These findings highlight the complex regulation of iron transport in FGR placentae, suggesting a selective downregulation of mitoferrin-2 while pre-serving mitoferrin-1 iron uptake for haematopoietic use.

### Altered Fe-S cluster biosynthesis machinery in FGR placental mitochondria

Given that our data suggested a conservation of intra-cellular iron and altered mitochondrial iron uptake within FGR placentae, we went on to investigate Fe-S cluster biosynthesis. Fe-S clusters are apoproteins essential for cellular activities, including haem, and are directly incorporated into the mitochondrial ETC to support mitochondrial structure and function (Guan et al., 2022; Voltarelli et al., 2023). Fe-S assembly is a complex process that can be categorised into two main components: (i) *de novo* [2Fe-2S] cluster synthesis and (ii) late Fe-S ([4Fe-4S]) cluster synthesis.

## Mitochondrial *de novo* Fe-S synthesis in FGR

*De novo* Fe-S cluster synthesis is a complex series of reactions, initiated by the catalysis of a pyridoxal-phosphate, cysteine desulfurase NFS1 (Maio & Rouault, 2020). Stabilising the activity of NFS1, NADH: ubiquinone oxidoreductase subunit AB1 (NDUFAB1) forms a heterodimer with LYR motif containing 4 (LYRM4), initiating Fe-S cluster synthesis via the complex NFS1:LYRM4:NDUFBA1 (Herrera et al., 2019). Our data showed a lower abundance of NDUFAB1 in FGR placentae, despite LYRM4 mRNA expression remaining unchanged. The subsequent synthesis of the NFS1:LYRM4:NDUFBA1 complex results in the binding of the iron–sulphur cluster assembly enzyme (ISCU) and frataxin (FXN), forming a hetero-octamer that consists of two copies of the NFS1:LYRM4:NDUFBA1:FXN:ISCU complex (Maio & Rouault, 2020).

In our study, FXN mRNA expression was similar between the two groups, suggesting any deficits in Fe-S assembly probably occur downstream of initial FXN-mediated iron and sulphide delivery onto ISCU, as loss of FXN results in widespread depletion of Fe-containing proteins and attenuation of mitochondrial protein synthesis (Zhong et al., 2023). We observed no change in ISCU mRNA, which provides the structural scaffold for [2Fe-2S] formation, in FGR placental mitochondria. However, this observation was not conserved in our proteomics data, with a higher protein abundance in FGR. In addition, key [2Fe-2S] proteins ferredoxin reductase (FDXR) and ferredoxin 2 (FDX2), which are required to supply electrons to complete [2Fe-2S] and [4Fe-4S] assembly, had lower protein and mRNA expression in FGR placentae, respectively. We therefore suggest that the FGR placenta prioritises [2Fe-2S] biogenesis at the detriment of [4Fe-4S] cluster synthesis. This is consistent with previous knockdown studies showing that loss of FDX2 globally disrupts mitochondrial Fe-S assembly, leading to dysregulated oxidative phosphorylation and reduced antioxidant production (Shi et al., 2012).

Following [2Fe-2S] cluster formation, GLRX5 receives the [2Fe-2S] cluster from ISCU for transport to apoproteins required for various cellular and mitochondrial functions (Paul et al., 2017). We observed a significant decrease in mRNA expression of GLRX5, yet GLRX5 protein abundance remained unchanged between the two groups, again preserving the capacity for [2Fe-2S] cluster transfer to downstream cellular and mitochondrial functions such as haem synthesis or late-stage Fe-S cluster formation.

## Mitochondrial late-stage Fe-S synthesis in FGR

Given the changes observed in FDXR and FDX2 and their sequential roles in the formation of [2Fe-2S] and [4Fe-4S] clusters, we next investigated late-stage Fe-S synthesis. Late-stage [4Fe-4S] clusters are critical cofactors that are incorporated into mitochondrial ETC complexes, and apoproteins such as iron-regulatory proteins, and anti-oxidants (Maio & Rouault, 2020).

The formation of [4Fe-4S] clusters requires the transfer of [2Fe-2S] intermediaries via GLRX5 to the A-type scaffold proteins, iron–sulphur cluster assembly 1 and 2 (ISCA1 and ISCA2), in conjunction with iron–sulphur cluster assembly factor IBA57 (Paul et al., 2017). In FGR placentae, we observed a small increase in IBA57 protein, which was confirmed by western blotting. However, we did not detect ISCA1 and ISCA2 at sufficient levels within our proteomic analysis. Given that ISCA1 and ISCA2 are known to contain [2Fe-2S] clusters (Beilschmidt et al., 2017) but IBA57 cannot alone bind [2Fe-2S] clusters (Nasta et al., 2019), the inability to detect sufficient levels of ISCA proteins in FGR may suggest that the newly synthesised [2Fe-2S] clusters cannot be effectively matured into [4Fe-4S] clusters.

Newly synthesised [4Fe-4S] clusters are released from the ISCA1:ISCA2:IBA57 complex by a chaperone/co-chaperone system comprising heat shock protein family A member 9 (HSPA9) and mitochondrial Fe-S cluster co-chaperone (HSC20). HSPA9 protein abundance was lower in FGR placentae. HSPA9 mediates the transfer [4Fe-4S] clusters directly to apoproteins or through targeting factors such as NFU1 iron–sulphur cluster scaffold (NFU1), BolA family member 3 (BOLA3) and NUBP iron–sulphur cluster assembly factor (NUBPL) (Maio & Rouault, 2020; Melber et al., 2016; Zhong et al., 2023). Despite a significant decrease in NUBPL mRNA expression in FGR placental tissue (Fig. 2*K*), we observed increased mRNA expression of BOLA3 and NFU1 in FGR placentae (Fig. 2*I* and *J*), which may suggest an attempt to maximise the delivery of a limited pool of [4Fe-4S] clusters.

Collectively, the reduced FDX2 mRNA expression and lower abundance of FDXR and HSPA9 suggest a bottleneck in mitochondrial Fe-S cluster biogenesis in the FGR placenta. While these proteins participate in both [2Fe-2S] and [4Fe-4S] cluster formation (Maio & Rouault, 2020), their decreased abundance in FGR likely prioritises the direction of limited resources towards [2Fe-2S] cluster formation. This disruption in mitochondrial Fe-S assembly may provide a molecular basis for the mitochondrial dysfunction observed in FGR placentae,

characterised by impaired respiratory chain complex activity, altered ATP production and increased oxidative stress (Holland et al., 2017; Xu et al., 2021).

## Altered haem synthesis and erythrocyte structure in FGR placentae: implications for fetal oxygenation and development

Based on our data showing prioritisation of [2Fe-2S] clusters in FGR placentae, we further investigated the haem synthesis pathway. Haem synthesis critically depends on the insertion of [2Fe-2S] clusters into ferrochelatase (FECH), the terminal enzyme of haem biosynthesis that resides within the mitochondria. The insertion of [2Fe-2S] clusters into FECH ensures its stability and activity for haem production (Obi et al., 2022). Although our data revealed lower mRNA expression of coproporphyrinogen oxidase (CPOX), protein levels of CPOX were higher in FGR placental tissue. Additionally, elevated mRNA and protein levels of FECH, located within the mitochondria, were accompanied by greater protein abundance of haem enzymes hydroxymethylbilane synthase (HMBS) and uroporphyrinogen decarboxylase (UROD). These data support our suggestion that the FGR placentae prioritise [2Fe-2S] clusters and shift utilisation towards haem synthesis. Additionally, FECH has been reported to be complexed to ATP binding cassette subfamily B member 10 (ABCB10), conferring stability to mitoferrin-1. This FECH–ABCB10 conjugation may explain why mitoferrin-1 expression remains conserved in FGR placentae despite reduced mitoferrin-2.

Beyond its role in oxygen transport and haemoglobin production, haem also contributes to mitochondrial respiration by facilitating electron transfer and proton translocation within ETC complexes II, III and IV through its incorporation as haem a, b and c moieties (Kim et al., 2012). Despite the increased levels of haem synthesis enzymes, the assessment of haem concentration in FGR placentae was lower. This may be due to increased incorporation of haem a, b and c into ETC complexes II, III and IV to increase mitochondrial respiration resulting from a lack of [4Fe-4S] assembly, although this hypothesis requires confirmation.

Alternatively, haem is a known pro-oxidant, inducing oxidative stress, and haem accumulation can trigger its catabolism via haem oxygenase 1 (HMOX1), which degrades haem into carbon monoxide, iron and biliverdin. We observed greater HMOX1 abundance in FGR placentae, suggesting that despite upregulation of the haem biosynthesis pathway, increased haem breakdown may reflect heightened oxidative stress or a compensatory mechanism to regulate 'free' haem levels. The catabolism of haem by the placenta is not well understood (Liu et al.,

2016), although Peoc'h et al. (2020) have suggested that HMOX expression in the placenta could be a protective response against oxidative stress, which may explain why HMOX1 was higher in FGR placentae.

Our findings suggest a delicate balance in placental haem metabolism in FGR, whereby the upregulation of haem synthesis pathways likely represents an adaptive response to fetoplacental hypoxia and maintenance of placental bioenergetics. Simultaneously, the increase in HMOX1-mediated haem catabolism serves as a protective mechanism against haem-induced oxidative stress.

## Haematopoietic function in FGR placentae

In addition to its role in mitochondrial function, haem is integral for the production of various globin chains, including gamma ($\gamma$), alpha ($\alpha$) and beta ($\beta$). This fundamental process is central to the oxygen-carrying capacity of blood and, by extension, aerobic respiration essential for cellular function and viability (Hamza & Dailey, 2012; Voltarelli et al., 2023). Our findings revealed lower protein abundance of gamma-1 (HBG1) and gamma-2 (HBG2), key beta-like globin components of fetal haemoglobin, along with haemoglobin subunit alpha 1 (HBA1), in FGR placentae compared to healthy term placentae. Additionally, the mRNA expression of haemoglobin subunit beta (HBB), a critical component of adult haemoglobin ($\alpha_2\beta_2$), was significantly greater in FGR placentae, however, HBB protein abundance was lower in FGR placentae. The presence of HBB in the placenta is consistent with the developmental switch to adult $\beta$-globin expression, which begins during the third trimester, as the primary site of erythropoiesis transitions from the fetal liver to the bone marrow (Cantú & Philipsen, 2014; Khandros & Blobel, 2024).

While fetal hypoxia in FGR is typically associated with increased fetal haemoglobin synthesis (Chang et al., 2018), our findings indicate a disruption in globin chain production. The lower protein abundance of key globin subunits, HBG1, HBG2, HBB and HBA1, in FGR placentae suggests that synthesis or stability is impaired, potentially limiting the assembly of functional haemoglobin. Although recent studies describe the placenta as a transient haematopoietic niche (Ivanovs et al., 2017; Rhodes et al., 2008), this function may be compromised in FGR. Chronic hypoxia, inflammation and oxidative stress, all hallmarks of FGR pathophysiology, can disrupt haematopoietic stem and progenitor cell function and erythroid differentiation (Dzierzak & Bigas, 2018). While hypoxia is typically associated with the upregulation of fetal haemoglobin (Khandros & Blobel, 2024), our findings revealed lower levels of both fetal and adult haemoglobin subunits in the FGR placentae. Collectively, these findings suggest that

there is a dysfunctional erythropoietic environment in the FGR placenta.

Given the substantial alterations in both haem and globin subunits, we further investigated the erythropoietic structural proteins in FGR placentae. Notably, we observed reductions in erythrocyte membrane protein band 4.1 (EPB41) expression at both the mRNA and protein levels in FGR placentae. EPB41 plays a crucial role in stabilising the actin–spectrin cytoskeleton of erythrocytes, thereby maintaining the characteristic biconcave disc shape of erythrocytes, which is fundamental for facilitating efficient oxygen transport (Uzoigwe, 2006; Vorn et al., 2023). Furthermore, we observed reduced protein abundance in EPB42, SPTA1, SPTB, ANK1 and SLC4A1 in FGR placentae when compared to healthy controls. Similarly, these proteins are integral components of the erythrocyte membrane, conferring flexibility and stability (Alper, 2009; Guo et al., 2022). Our findings suggest altered erythropoietic structure during erythrocyte maturation, as HEMGN mRNA expression remained unchanged, indicating that the HSPC population and early erythroid commitment remain preserved. Haemogen regulates the proliferation and differentiation of haematopoietic cells (Guo et al., 2022), indicating that the architectural insult observed in FGR placentae is likely not occurring during haematopoiesis but rather during erythrocyte maturation (Vorn et al., 2023) or as a result of stress-induced erythrocyte maturation (Paulson et al., 2020). Collectively, these alterations may represent an attempt to increase erythrocyte deformability to facilitate circulation within the structurally compromised vasculature of the FGR placenta, thus enabling efficient uptake and transport of the limited oxygen available to the fetus.

## Limitations

While maternal iron status is a critical determinant of fetal iron acquisition through placental transport (Roberts et al., 2020), our study revealed intrinsic placental changes independent of maternal iron status, revealing potential intracellular alterations that have remained elusive in the development of FGR. Since the placenta is the critical interface between maternal and fetal circulations, future studies incorporating fetal cord blood analysis would provide further insight into how placental modifications impact placental iron transport and fetal iron status.

This study employed a bottom-up LC-MS proteomic approach to identify alterations in placental iron transport and mitochondrial iron utilisation. While this methodology is well-suited for global cellular protein profiling, the detection of some proteins were difficult due to low-abundance, membrane-bound proteins and incomplete coverage of post-translational modifications (Bruderer et al., 2015; Mulhall et al., 2024). Notably,

Fe-S cluster biosynthesis proteins were difficult to detect, likely due to their hydrophobic nature and limitations of the methodology. Future studies incorporating targeted approaches may improve the detection and quantification of these proteins.

## Conclusion

Our study reveals specific alterations in placental iron transport and utilisation pathways in FGR that are independent of maternal iron status. We identified increased placental iron transporters alongside reduced ferroportin, suggesting adaptive iron retention within FGR placentae. Assessment of mitochondrial iron utilisation pathways in FGR placentae suggested a preference for [2Fe-2S] cluster formation and haem synthesis over [4Fe-4S] cluster assembly in FGR. Collectively, these findings of altered placental iron transporters and modifications in mitochondrial Fe-S cluster and haem synthesis pathways likely reflect the placenta's response to the compromised vascular environment characteristic of FGR. We also observed reduced haemoglobin subunits and changes in erythrocyte structural proteins in FGR placentae, indicative of altered placental haematopoietic function. We propose these data are a programmed response to hypoxia and stress-induced haematopoiesis to overcome the poor placental vasculature in FGR. Together, these findings provide new insights into how alterations in iron transport and iron-dependent pathways contribute to the development of FGR.

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

## Additional information

### Data availability statement

The authors will make the raw data supporting the conclusions of this article available, without undue reservation, to any qualified researcher.

### Competing interests

The authors declare that they have no conflict of interest relating to this study.

### Author contributions

V.B.B., K.G.P., R.S. and J.J.F. contributed to the conception and design of the study. V.B.B., H.M. and S.A. contributed to the acquisition, analysis and interpretation of data for the study. V.B.B., K.G.P., R.S. and J.J.F. contributed to drafting the work and critically revising the intellectual content. All authors have approved the final version of the manuscript. All authors agree to be accountable for all aspects of the work. All persons designated as authors qualify for authorship and are listed. This manuscript was first published as a preprint: Veronica B. Botha, Heather C. Murray, Siddharth Acharya, Kirsty G. Pringle, Roger Smith, Joshua J. Fisher (2025). Placental utilisation in fetal growth restriction: alterations in mitochondrial heme synthesis and iron-sulfur cluster assembly pathway. BIORXIV/2025/658195. https://doi.org/10.1101/2025.06.09.658195.

### Funding

This research was supported by the National Health and Medical Research Council Investigator Grant GNT2026065.

### Acknowledgements

We gratefully acknowledge the research midwives at John Hunter Hospital, Skye Doel and Bridget McCleery, who were involved in participant recruitment and clinical data collection, for their invaluable support and contribution to this study.

Open access publishing facilitated by The University of Newcastle, as part of the Wiley - The University of Newcastle agreement via the Council of Australian University Librarians.

### Keywords

erythrocyte structure, fetal growth restriction, haem synthesis, iron transport, iron–sulphur clusters, mitochondria, placenta

### Supporting information

Additional supporting information can be found online in the Supporting Information section at the end of the HTML view of the article. Supporting information files available:

**Peer Review History**

