## [Peer Review History · The Journal of Physiology]

Placental Iron Utilisation in Fetal Growth Restriction: Alterations in Mitochondrial Heme Synthesis and Iron-Sulfur Cluster Assembly Pathways

Veronica B Botha, Heather C Murray, Siddharth Acharya, Kirsty G Pringle, Roger Smith, and Joshua J Fisher
DOI: 10.1113/JP289451

Corresponding author(s): Joshua Fisher (joshua.fisher@newcastle.edu.au)

Review Timeline:

Submission Date:	16-Jun-2025
Editorial Decision:	28-Jul-2025
Revision Received:	09-Oct-2025
Editorial Decision:	27-Oct-2025
Revision Received:	25-Nov-2025
Editorial Decision:	16-Dec-2025
Revision Received:	08-Jan-2026
Accepted:	14-Jan-2026

Senior Editor: Laura Bennet

Reviewing Editor: Janna Morrison

Transaction Report:

Dear Dr Fisher,

Re: JP-RP-2025-289451 "**Placental Iron Utilisation in Fetal Growth Restriction: Alterations in Mitochondrial Heme Synthesis and Iron-Sulfur Cluster Assembly Pathways**" by Veronica B Botha, Heather C Murray, Siddharth Acharya, Kirsty G Pringle, Roger Smith, and Joshua J Fisher

Thank you for submitting your manuscript to The Journal of Physiology. It has been assessed by a Reviewing Editor and by 1 expert referees and we are pleased to tell you that it is acceptable for publication following satisfactory revision.

REVISION CHECKLIST:

We look forward to receiving your revised submission.

Yours sincerely,

Laura Bennet
Senior Editor
The Journal of Physiology

REQUIRED ITEMS

- 1)- Author photo and profile. First or joint first authors are asked to provide a short biography (no more than 100 words for one author or 150 words in total for joint first authors) and a portrait photograph. These should be uploaded and clearly labelled together in a Word document with the revised version of the manuscript. See Information for Authors for further details.
- 2)- The contact information for the person responsible for 'Research Governance' at your institution needs to be provided. This includes their name and an institutional email address. Please ensure the contact is not an author on this paper and provide an alternate contact if necessary, or confirm in the submission form that the author whose email was provided has sole responsibility for research governance. This is the person who is responsible for regulations, principles and standards of good practice in research carried out at the institution, for instance the ethical treatment of animals, the keeping of proper experimental records or the reporting of results.
- 3)- You must start the Methods section with a paragraph headed Ethical Approval. If experiments were conducted on humans, confirmation that informed consent was obtained, preferably in writing, that the studies conformed to the standards set by the latest revision of the Declaration of Helsinki and that the procedures were approved by a properly constituted ethics committee, which should be named, must be included in the article file. If the research study was registered (clause 35 of the Declaration of Helsinki), the registration database should be indicated, otherwise the lack of registration should be noted as an exception (e.g. The study conformed to the standards set by the Declaration of Helsinki, except for registration in a database). For further information see: <https://physoc.onlinelibrary.wiley.com/hub/human-experiments>.
- 4)- Your manuscript must include a complete Additional Information section, with the heading "Additional Information", including competing interests; funding; author contributions and acknowledgements.
- 5)- Please upload separate high-quality figure files via the submission form.
- 6)- Papers must comply with the Statistics Policy: https://jp.msubmit.net/cgi-bin/main.plex?form_type=display_requirements#statistics.

In summary:

- If n {less than or equal to} 30, all data points must be plotted in the figure in a way that reveals their range and distribution. A bar graph with data points overlaid, a box and whisker plot or a violin plot (preferably with data points included) are acceptable formats.
- If $n > 30$, then the entire raw dataset must be made available either as supporting information, or hosted on a not-for-profit repository, e.g. FigShare, with access details provided in the manuscript.

- 'n' clearly defined (e.g. x cells from y slices in z animals) in the Methods. Authors should be mindful of pseudoreplication.
- All relevant 'n' values must be clearly stated in the main text, figures and tables.
- The most appropriate summary statistic (e.g. mean or median and standard deviation) must be used. Standard Error of the Mean (SEM) alone is not permitted.
- Exact p values must be stated. Authors must not use 'greater than' or 'less than'. Exact p values must be stated to three significant figures even when 'no statistical significance' is claimed.

7) - Please include an Abstract Figure file, as well as the Figure Legend text within the main article file. The Abstract Figure is a piece of artwork designed to give readers an immediate understanding of the research and should summarise the main conclusions. If possible, the image should be easily 'readable' from left to right or top to bottom. It should show the physiological relevance of the manuscript so readers can assess the importance and content of its findings. Abstract Figures should not merely recapitulate other figures in the manuscript. Please try to keep the diagram as simple as possible and without superfluous information that may distract from the main conclusion(s). Abstract Figures must be provided by authors no later than the revised manuscript stage and should be uploaded as a separate file during online submission labelled as File Type 'Abstract Figure'. Please also ensure that you include the figure legend in the main article file. All Abstract Figures should be created using BioRender. Authors should use The Journal's premium BioRender account to export high-resolution images. Details on how to use and access the premium account are included as part of this email.

8) - Please ensure that all figures and tables have a title and legend, and that they have been cited within the main article text.

EDITOR COMMENTS

Reviewing Editor:

Thank you for submitting this manuscript. Please revise the paper based on the reviewers comments.

Senior Editor:

Thank you for your submission. When revising the manuscript, please keep in mind The Journal guidelines for statistics. The reviewer has highlighted a few areas to address and some data are in SEM

REFEREE COMMENTS

Referee #2:

The study by Botha, et al. compared iron utilisation in placentas from term, uncomplicated pregnancies and pregnancies complicated by FGR. This observational study provides new insight into placental mechanisms of heme and iron-sulphur cluster synthesis and how FGR may impact these processes. Despite the detailed molecular assessment, there are some considerations regarding the study and its interpretation that are required.

Introduction

Overall, this is well written; however, lines 74-75 reads as though inadequate uterine spiral artery remodelling is the only uteroplacental cause of FGR. The authors should acknowledge other uteroplacental insults that can contribute to FGR.

Methodology

Line 182-183: What was the mode of delivery for the healthy, term pregnancies?

Lines 185-188: Is there information regarding patient supplement use and/or dietary habits?

Line 263: What was the reasoning for including only a sub-set of placental tissue for protein analysis by WB and what was the sex distribution of placentas within this subset?

Line 291: Why were both Grubbs and ROUT tests used to identify outliers?

Lines 295-296: Data presented should be mean{plus minus}SD, as per author guidelines.

Lines 302-306: What was the statistical significance threshold for the Proteomic analysis as it does not seem to be stated. Why was there no log2FC threshold determined (e.g., were there any predetermined parameters established to determine "biologically significant" results)? For example, a $|\log_2FC|$ of <0.5 may be significant, but its biological significance is questionable. The data analysed appears to have used FGR as the reference group (i.e., a positive log2FC indicates expression is higher in control vs FGR) - it would make more sense to use the control group as the reference group, and this would resolve some confusion when interpreting the data (see below).

Results

A general comment - while the authors have not considered placental sex as a variable in their analysis (likely due to small sample sizes), it would benefit all graphs to assign different symbols for female and male samples.

Lines 313-314: While the authors state that gestational age at delivery and mode of delivery were included in analysis, this data is not shown in Table 2.

Lines 323-325: As this data is not significant, it should be removed as per Statistics policy of Author Guidelines.

Line 353: Figure 1D - the data is described as expression being lower in FGR but is supported by a positive log2FC - this is counterintuitive. The proteomic data would be more informative to describe the change of direction in FGR relative to term control (i.e., "FPN protein analysis... lower protein levels (-0.646 log2FC)"). The legend for graph 1D (and subsequent heatmap data) should be changed to reflect this.

Lines 357-359: What was the sex distribution of placentas used for histological analysis?

Discussion

The discussion is detailed and addresses all key findings from the study; however, it is too long and should be edited down.

Considering the role of iron in mitochondrial function, it would strengthen the study to assess mitochondrial abundance and/or function/expression of ETC complexes within this cohort.

Further comments on the clinical implications of the findings would strengthen the discussion.

It is unclear why the authors switched from log2FC to FC in protein expression when discussing the results of their study. Please clarify.

Line 818: sentence ends abruptly.

END OF COMMENTS

Referee #2:

Introduction:

Overall, this is well written; however, lines 74-75 reads as though inadequate uterine spiral artery remodelling is the only uteroplacental cause of FGR. The authors should acknowledge other uteroplacental insults that can contribute to FGR.

Response:

We thank the reviewer for their comment. We agree that the original sentence may have implied that inadequate spiral artery remodelling is the sole uteroplacental cause of FGR. To address this, we have revised the sentence to acknowledge additional uteroplacental insults, including impaired trophoblast invasion, abnormal vascular development, oxidative stress, and inflammation.

“Placental insufficiency can arise from inadequate uterine spiral artery remodelling, impaired trophoblast invasion, abnormal vascular development, oxidative stress and inflammation (Brooker *et al.*, 2025; Higashijima *et al.*, 2013; Tang *et al.*, 2017).”
(Lines 84-87).

This change is supported by the cited literature by Brooker *et al.*, 2025; Higashijima *et al.*, 2013; Tang *et al.*, 2017.

Methodology

Line 182-183: What was the mode of delivery for the healthy, term pregnancies?

Response:

We have included this information in the Methods section of the revised manuscript, Line 192-194: “All placentae collected for this study were healthy term placentae from singleton pregnancies delivered between 37-40 weeks of gestation either vaginally or by caesarean section.”

Lines 185-188: Is there information regarding patient supplement use and/or dietary habits?

Response:

We appreciate the reviewer's question. While we acknowledge that dietary habits and supplement use inherently play a role in overall health during pregnancy, monitoring these factors would require a prospective study. As our study was retrospective, dietary habits and supplementation tracking were outside the scope of

this manuscript. However, as noted in the manuscript, we did assess maternal circulating iron, and observed no statistical changes (Table 2).

Line 263: What was the reasoning for including only a sub-set of placental tissue for protein analysis by WB and what was the sex distribution of placentas within this subset?

Response:

The rationale for performing WB on a subset of placental tissue was due to cost and tissue availability. Importantly, our subset had adequate power to detect significance as evidenced by Figure 2 (M). Furthermore, the WB was run in support of our proteomics data.

Overall, there were 7 placentae per group, which reflected the broader sample demographic in terms of sex (Control: n = 4 males, n = 3 females; FGR: n = 3 males, n = 4 females), gestational age, and FGR pathology severity (<3rd, <5th, <10th). This information has now been included in the methods: "To validate our proteomics data, western blotting was performed on the same subset of healthy (n = 7; 4 males, 3 females) and FGR (n=7; 3 males, 4 females, n=3 <3rd, n=1 <5th, n=3 <10th)." (Lines 279-281)

Line 291: Why were both Grubbs and ROUT tests used to identify outliers?

Response: We appreciate the reviewer's question regarding our use of multiple approaches for outlier detection. Both Grubbs and ROUT tests were used together, as previous studies have shown that using both methods enhances sensitivity and reliability in outlier detection (Klingelhöfer, 2023). The combined use of both methods provides a rigorous approach to outlier identification, ensuring statistical validity and transparency. We believe this produces the most thorough and reproducible dataset for future studies. We have now clarified this rationale in the Methods section, Lines 310-316: "Outlier detection was performed using the ROUT test due to its robustness in detecting multiple outliers while controlling the false discovery rate (Motulsky & Brown, 2006). Subsequent Grubbs testing ($\alpha=0.05$) validated values and identified additional outliers under normality assumptions (Grubbs, 1969) while concurrently flagging data points with z scores exceeding ± 2 standard deviations from the group mean (Andrade, 2021). These data points were excluded to account for human variability (Andrade, 2021).

Lines 295-296: Data presented should be mean {plus minus} SD, as per author guidelines.

Response:

We thank the reviewer for bringing this to our attention. All figures and legends have been revised to present the "mean \pm standard deviation (SD)" (lines 323-324), in accordance with the journal's guidelines of the methods section and figure legends.

Lines 302-306: What was the statistical significance threshold for the Proteomic analysis as it does not seem to be stated. Why was there no log₂FC threshold determined (e.g., were there any predetermined parameters established to determine "biologically significant" results)? For example, a |log₂FC| of <0.5 may be significant, but its biological significance is questionable. The data analysed appears to have used FGR as the reference group (i.e., a positive log₂FC indicates expression is higher in control vs FGR) - it would make more sense to use the control group as the reference group, and this would resolve some confusion when interpreting the data (see below).

Response:

We appreciate the opportunity to clarify our proteomic analysis. Our analysis did not aim to address the most significant differences between healthy and FGR placentas, as a more traditional proteomics manuscript would. Indeed, we focused on assessing differences in protein abundance between FGR and term control placentae within specific pathways of interest. Consequently, we did not apply a predefined log₂ fold change cutoff. We validated findings using western blotting, as noted in the manuscript (Line: 264). Control samples were used as the reference group; thus, positive values indicate lower expression in FGR compared to controls. We believe this misunderstanding may have arisen due to the choice of wording in the proteomic heatmaps within figures, "Higher in control" and "Higher in FGR"; we have subsequently changed this to read "Higher in FGR" and "Lower in FGR" in all figures. Similarly, we have clarified this in the relevant figure legends and Methods section of our manuscript, Lines: 270-274: "Proteomic analysis employed a bottom-up approach to profile protein abundance (Murray *et al.*, 2023). Log₂ FC thresholds were not applied, to ensure that specific protein changes within relevant proteins of interest were not overlooked or masked by larger magnitude changes within the

dataset, as previously published (Aebersold & Mann, 2016; Bandeira *et al.*, 2021; Geyer *et al.*, 2016)” and Lines 330-334: “The log₂ FC was derived from the ratio between healthy term placentae to assess changes in FGR placentae, as such a positive log₂ FC indicates lower protein abundance in FGR placentae, while a negative log₂ FC indicates higher protein abundance in FGR placentae. Proteins with fewer than 2 Peptide Spectrum Matches (PSMs) were excluded from the analysis to ensure data reliability.”

Results

A general comment - while the authors have not considered placental sex as a variable in their analysis (likely due to small sample sizes), it would benefit all graphs to assign different symbols for female and male samples.

Response:

As the Reviewer suggested, while placental sex was not considered a variable due to the small sample size, we have modified the graphs to reflect the sex distribution within the datasets. Females have been assigned open shapes while males have been assigned filled shapes. We have included this distinction within Lines 308-310 of the methods section and figure legends: “Analysis of fetal sex was not conducted due to insufficient sample size, however, for transparency, female data points are represented by open shapes in the figures, while closed shapes represent male data points.”

Lines 313-314: While the authors state that gestational age at delivery and mode of delivery were included in the analysis, this data is not shown in Table 2.

Response:

We apologise for the error, gestational age at delivery and mode of delivery were not included in Table 2 but are described in text lines 192-196. This section has been amended and now reads: “Analysis included maternal parameters (age and BMI), birth outcomes (birth weight, birth centile, placental weight, fetal sex) and fetal biometry measurements following sonography assessment (estimated fetal weight (EFW%), abdominal circumference, femur length, head circumference, and amniotic fluid volume).” (Line 340-343)

Lines 323-325: As this data is not significant, it should be removed as per Statistics policy of Author Guidelines.

Response:

We thank the reviewer for their comment regarding the inclusion of non-significant data. We respectfully note that the Journal of Physiology's statistics policy does not require the removal of non-significant results. Instead, it promotes transparent reporting of all relevant findings, including those that are not statistically significant, as long as they are appropriately contextualised. In line with this, we have retained non-significant data that contribute to the overall interpretation of our results in FGR versus control placentae. These findings are significant as they establish that maternal haematological indices in our dataset is not a contributing component in the development of FGR pathology within our study.

Line 353: Figure 1D - the data is described as expression being lower in FGR but is supported by a positive log₂FC - this is counterintuitive. The proteomic data would be more informative to describe the change of direction in FGR relative to term control (i.e., "FPN protein analysis... lower protein levels (-0.641 log₂FC)"). The legend for graph 1D (and subsequent heatmap data) should be changed to reflect this.

Response:

We apologise for this confusion. As described above, "The log₂ FC was derived from the ratio between healthy term placentae to assess changes in FGR placentae, so a negative log₂ FC indicates higher protein abundance in FGR placentae (blue), and a positive log₂ FC indicates lower protein abundance in FGR placentae (orange)" (Lines 398-401).

Lines 357-359: What was the sex distribution of placentas used for histological analysis?

Response:

The same subset of placental tissue used for western blotting was used to for histological analysis. This has now been clarified in methods section lines 207-208 "...healthy (n = 7; 4 males, 3 females) and FGR (n=7; 3 males, 4 females, n=3 <3rd, n=1 <5th, n=3 <10th) placental tissue..." and Lines 384-387 "Histological analysis by

Prussian blue staining, using the same subset of control (n = 7; 4 males, 3 females) and FGR (n=7; 3 males, 4 females, n=3 <3rd, n=1 <5th, n=3 <10th) placentae analysed for western blotting and proteomics...”

Discussion

The discussion is detailed and addresses all key findings from the study; however, it is too long and should be edited down.

Response:

We appreciate the reviewer's suggestion regarding the length of the discussion.

Given the complexity of the study, which encompasses 4 key interconnected pathways, and the importance of addressing our key findings, we believe that further condensation could risk oversimplifying and potentially omit important context, leading to misinterpretation of results in future citations.

Considering the role of iron in mitochondrial function, it would strengthen the study to assess mitochondrial abundance and/or function/expression of ETC complexes within this cohort.

Response:

We thank the reviewer for this suggestion and agree that assessing mitochondrial abundance and/or function of electron transport chain complexes would provide valuable mechanistic insight into the role of iron in mitochondrial function within the FGR placentae. However, due to the retrospective nature of this study, functional assessments on mitochondrial respiration were not performed as they require fresh tissue, or adequately prepared samples at time of collection.

Mitochondrial abundance is commonly estimated via the mtDNA:nDNA ratio (mitochondrial content), which has been validated in various tissues using qPCR (Quiros et al., 2017; Ajaz et al., 2015), however, selecting appropriate genes to derive this ratio is critical for producing reproducible and accurate data. Mitochondrial abundance was not assessed in this manuscript as FGR, has been shown to alter both nuclear and mitochondrial encoded genes (Kiyokoba et al., 2022(Hu *et al.*, 2024; Naha *et al.*, 2020; Xu *et al.*, 2021)). As such, determining which mitochondrial and nuclear gene to derive this ratio may influence the ratio and may lead to misinterpretation. In addition, it was not the intention of this manuscript to examine mitochondrial networks in FGR placenta. Furthermore, abundance does not

reflect function, as dysfunctional mitochondria may retain or even increase their mitochondrial content (Yen et al., 2024).

Further comments on the clinical implications of the findings would strengthen the discussion.

Response:

We have added to the discussion the clinical implications that our findings highlight.

Lines 596-598: However, RDW should be interpreted cautiously in the presence of inflammation (Owoicho et al. 2022; Fava et al. 2019).

Lines 628-631: These findings support the need to reassess current diagnostic criteria (Beard 2000). As supported by Rusch et al. (2023), the diagnosis of iron deficiency remains complex due to limitations in traditional markers of iron status and should be interpreted in relation to the patient's overall health.

It is unclear why the authors switched from log₂FC to FC in protein expression when discussing the results of their study. Please clarify.

Response:

We apologise for the confusion. To improve clarity and consistency, we have revised the manuscript to use log₂ FC exclusively throughout all sections, including the discussion, figure legends, and tables.

Line 818: sentence ends abruptly.

Response:

We thank the reviewer for this comment and have modified the sentence for better flow and clarity (Lines 864-868): "While hypoxia is typically associated with the upregulation of fetal haemoglobin via HIF1 α activation (Khandros & Blobel, 2024), our findings revealed lower levels of both fetal and adult haemoglobin subunits in the FGR placentae. Collectively, these findings suggest that there may be a dysfunctional erythropoietic environment in the FGR placenta."

Dear Dr Fisher,

Re: JP-RP-2025-289451R1 "**Placental Iron Utilisation in Fetal Growth Restriction: Alterations in Mitochondrial Heme Synthesis and Iron-Sulfur Cluster Assembly Pathways**" by Veronica B Botha, Heather C Murray, Siddharth Acharya, Kirsty G Pringle, Roger Smith, and Joshua J Fisher

Thank you for submitting your manuscript to The Journal of Physiology. It has been assessed by a Reviewing Editor and by 1 expert referees and we are pleased to tell you that it is acceptable for publication following satisfactory revision.

REVISION CHECKLIST:

Please upload two versions of your manuscript text: one with all relevant changes highlighted and one clean version with no changes tracked. The manuscript file should include all tables and figure legends, but each figure/graph should be uploaded as separate, high-resolution files. The journal is now integrated with Wiley's Image Checking service. For further details, see: <https://www.wiley.com/en-us/network/publishing/research-publishing/trending-stories/upholding-image-integrity-wileys->

image-screening-service

We look forward to receiving your revised submission.

Yours sincerely,

Laura Bennet
Senior Editor
The Journal of Physiology

EDITOR COMMENTS

Reviewing Editor:

Thank you for revising the paper.

Please note that the sample size should be indicated in each table and figure legend.

Please clarify the approach to statistical analysis.

REFEREE COMMENTS

Referee #2:

I thank the authors for their detailed response to the previous comments. I have two additional points that should be addressed.

Regarding the outlier analysis - the wording in lines 310-316 is not clear as to whether the same outliers were removed using both Grubbs and ROUT tests. Specifically, the authors state that "Subsequent Grubbs testing ($\alpha=0.05$) validated values and identified additional outliers under normality assumptions" on data that had previously been tested for outliers using ROUT testing; however, it is my understanding that Grubbs testing can only identify one outlier in a dataset. Does this mean that additional outliers, as per the authors' wording, were identified using Grubbs and removed after data was tested for outliers using ROUT? Further clarification is required. Can the authors please also provide additional details for the Klingelhöfer paper that they cite in their response to my previous comments?

Regarding the discussion - I am still of the opinion that a 12-page discussion is too long. While I appreciate that the authors' work is complex and encompasses multiple interconnected pathways - as many studies in the field do - the key messages and the significance of the work is ultimately lost in the large body of text. It is my recommendation that further refinement of the discussion is needed to ensure the key, take home messages of the work are clearer for readers of The Journal of Physiology.

END OF COMMENTS

Referee #2:

I thank the authors for their detailed response to the previous comments. I have two additional points that should be addressed.

Regarding the outlier analysis, the wording in lines 310-316 is not clear as to whether the same outliers were removed using both Grubbs and ROUT tests. Specifically, the authors state that "Subsequent Grubbs testing ($\alpha=0.05$) validated values and identified additional outliers under normality assumptions" on data that had previously been tested for outliers using ROUT testing; however, it is my understanding that Grubbs testing can only identify one outlier in a dataset. Does this mean that additional outliers, as per the authors' wording, were identified using Grubbs and removed after the data was tested for outliers using ROUT? Further clarification is required. Can the authors please also provide additional details for the Klingelhöfer paper that they cite in their response to my previous comments?

Response:

We thank the reviewer for requesting this important clarification. We have revised lines 310-315 to provide greater methodological transparency and address all concerns raised. We have also cited more robust publications than those provided in our previous iteration.

Lines 309-315: Outliers were identified using the ROUT test ($Q=1\%$), which detects multiple outliers while controlling the false discovery rate (Motulsky & Brown, 2006). The ROUT-cleaned dataset was subsequently analysed using Grubbs testing ($\alpha=0.05$) as a secondary validation under normality assumptions (Grubbs, 1969). Data points flagged by either method exhibited z-scores exceeding ± 2 standard deviations from the group mean and were excluded to account for human variation (Andrade, 2021). This approach was applied uniformly across all data sets.

We acknowledge the reviewer's point that the GRUBBS test identifies one outlier per dataset. In our dataset, Grubbs testing identified one additional outlier beyond those detected by ROUT. While ROUT is superior for datasets with multiple outliers (Motulsky and Brown 2006), published simulation data indicate that ROUT exhibits slightly higher rates of both false positives and false negatives when detecting single isolated outliers compared to Grubbs (GraphPad Prism Statistics Guide: https://www.graphpad.com/guides/prism/latest/statistics/stat_how_it_works_rout_method.htm). Our approach was conservative, intended to use GRUBBS as a secondary check under strict normality assumptions. This methodology was applied uniformly to all datasets prior to any group statistical analysis, ensuring consistency and minimising potential bias.

Regarding the discussion - I am still of the opinion that a 12-page discussion is too long. While I appreciate that the authors' work is complex and encompasses multiple interconnected pathways - as many studies in the field do - the key messages and

the significance of the work is ultimately lost in the large body of text. It is my recommendation that further refinement of the discussion is needed to ensure the key, take-home messages of the work are clearer for readers of The Journal of Physiology.

Response:

We acknowledge the reviewer's concern about the discussion length. In response, we have substantially condensed the discussion from 4,730 to 3,194 words to better align with The Journal of Physiology's readership expectations. The revised discussion now emphasises the key messages and the significance of our findings more clearly and concisely.

Dear Dr Fisher,

Re: JP-RP-2025-289451R2 **"Placental Iron Utilisation in Fetal Growth Restriction: Alterations in Mitochondrial Heme Synthesis and Iron-Sulfur Cluster Assembly Pathways "** by Veronica B Botha, Heather C Murray, Siddharth Acharya, Kirsty G Pringle, Roger Smith, and Joshua J Fisher

Thank you for submitting your manuscript to The Journal of Physiology. It has been assessed by a Reviewing Editor and by 1 expert referee and we are pleased to tell you that it is acceptable for publication following satisfactory revision.

REVISION CHECKLIST:

Please upload two versions of your manuscript text: one with all relevant changes highlighted and one clean version with no changes tracked. The manuscript file should include all tables and figure legends, but each figure/graph should be uploaded as separate, high-resolution files. The journal is now integrated with Wiley's Image Checking service. For further details, see: <https://www.wiley.com/en-us/network/publishing/research-publishing/trending-stories/upholding-image-integrity-wileys->

image-screening-service

We look forward to receiving your revised submission.

Yours sincerely,

Laura Bennet
Senior Editor
The Journal of Physiology

EDITOR COMMENTS

Reviewing Editor:

The reviewer has requested that the authors provide more detail about the statistics approach. Please provide the requested references or consider using only one approach to identify outliers.

REFEREE COMMENTS

Referee #2:

I would firstly like to thank the reviewers for refining the discussion. The take home messages from the manuscript are much clearer.

I also thank the authors for providing further clarification regarding statistical analysis, though I remain unconvinced that the chosen approach is appropriate. As the Grubbs test is intended to detect a single outlier in an unmodified and univariate dataset that follows an approximately normal distribution, previous outlier removal using ROUT would alter the distribution and thus break this assumption. For transparency and reproducibility, can the authors please include references to previous studies that have conducted Grubbs testing on a ROUT-cleaned dataset? If there are no studies that use this approach, I recommend using a single, pre-specified outlier approach (e.g., ROUT) and running all analyses on the ROUT-cleaned dataset. My main concern is that the current sequential approach may increase Type I error and bias the data towards normality. This is an important consideration for human datasets, which are often not normally distributed.

END OF COMMENTS

REFeree COMMENTS

Referee #2:

I would firstly like to thank the reviewers for refining the discussion. The take-home messages from the manuscript are much clearer.

I also thank the authors for providing further clarification regarding statistical analysis, though I remain unconvinced that the chosen approach is appropriate. As the Grubbs test is intended to detect a single outlier in an unmodified and univariate dataset that follows an approximately normal distribution, previous outlier removal using ROUT would alter the distribution and thus break this assumption. For transparency and reproducibility, can the authors please include references to previous studies that have conducted Grubbs testing on a ROUT-cleaned dataset? If there are no studies that use this approach, I recommend using a single, pre-specified outlier approach (e.g., ROUT) and running all analyses on the ROUT-cleaned dataset. My main concern is that the current sequential approach may increase Type I error and bias the data towards normality. This is an important consideration for human datasets, which are often not normally distributed.

Response:

Following the reviewer's recommendation, all data is now presented using a single, pre-specified outlier detection method (ROUT; $Q = 1\%$) applied uniformly across all datasets. No subsequent Grubbs or z-score-based filtering was performed.

This revised approach did not alter the overall direction or interpretation of findings related to placental iron transport or mitochondrial iron utilisation but did increase the variation within the data. Changes in significance were observed within Fe-S cluster pathways. Listed below

ISCU mRNA expression no longer reached statistical significance, however, ISCU protein abundance remained elevated in FGR placental tissue, supporting our previous discussion of preserved [2Fe-2S] cluster synthesis. Conversely, NUBPL mRNA expression was lower in FGR, further supporting our discussion and interpretation of the study as they stand, which highlighted constrained late-stage [4Fe-4S] cluster maturation and delivery (Figure 2).

Directionality of CPOX mRNA expression changed due to large variation, however, CPOX protein levels which were the foundation of our discussion remained higher in FGR placental tissue (Figure 4).

Collectively, the data continue to support our conclusion that mitochondrial iron utilisation in the FGR placenta is biased toward heme synthesis and [2Fe-2S] cluster formation, with constrained [4Fe-4S] cluster maturation. All relevant changes have been incorporated into the manuscript as detailed below.

Methods: Lines 319-321 now reads “Outliers were identified using the ROUT test (Q=1%), which detects multiple outliers while controlling the false discovery rate (Motulsky & Brown, 2006)”.

Results:

Lines 456–459 “Increased mRNA expression in [4Fe-4S] cluster transporters NFU1 Iron-Sulfur Cluster Scaffold (NFU1; $p=0.0120$; Figure 2I), BOLA Family member 3 (BOLA3; $p=0.0148$; Figure 2J), while NUBP Iron-Sulfur Cluster Assembly factor (NUBPL; $p=0.0118$; Figure 2K) mRNA expression was decreased”

Lines 486–490 “Analysis of the heme synthesis pathway showed significantly lower mRNA expression of coproporphyrinogen oxidase (CPOX; $p=0.002$; Figure 3A) in FGR placental villous tissue, yet protein levels were higher ($-0.163 \log_2$ FC; Figure 3D). In contrast, mRNA expression of ferrochelatase (FECH; Figure 3B) at the gene ($p=0.0036$) and protein level ($-0.356 \log_2$ FC; Figure 3D) were higher in FGR relative to controls”

Discussion:

Lines 678–679 “We observed no change in ISCU mRNA”,

Lines 720–723 “Despite a significant decrease in NUBPL mRNA expression in FGR placental tissue (Figure 2K), we observed increased mRNA expression of BOLA3 and NFU1 in FGR placentae (Figure 2I-J), which may suggest an attempt to maximise the delivery of a limited pool of [4Fe-4S] clusters.”

Lines 740–744 “Although our data revealed lower mRNA expression of coproporphyrinogen oxidase (CPOX), protein levels of CPOX were higher in FGR placental tissue. Additionally, elevated mRNA and protein levels of FECH...”

Dear Dr Fisher,

Re: JP-RP-2026-289451R3 "**Placental Iron Utilisation in Fetal Growth Restriction: Alterations in Mitochondrial Heme Synthesis and Iron-Sulfur Cluster Assembly Pathways**" by Veronica B Botha, Heather C Murray, Siddharth Acharya, Kirsty G Pringle, Roger Smith, and Joshua J Fisher

We are pleased to tell you that your paper has been accepted for publication in The Journal of Physiology.

Yours sincerely,

Laura Bennet
Senior Editor
The Journal of Physiology

IMPORTANT POINTS TO NOTE FOLLOWING ACCEPTANCE OF YOUR PAPER:

- **IMPORTANT NOTICE ABOUT OPEN ACCESS:** To assist authors whose funding agencies mandate immediate public access to published research findings, The Journal of Physiology allows authors to pay an Open Access (OA) fee to have their papers made freely available immediately on publication.

- You can help your research get the attention it deserves! Check out Wiley's free Promotion Guide for best-practice recommendations for promoting your work at: www.wileyauthors.com/eoo/guide. You can learn more about Wiley Editing Services which offers professional video, design, and writing services to create shareable video abstracts, infographics, conference posters, lay summaries, and research news stories for your research at: www.wileyauthors.com/eoo/promotion.

- If you would like to receive our 'Research Roundup', a monthly newsletter highlighting the cutting-edge research published in The Physiological Society's family of journals (The Journal of Physiology, Experimental Physiology, Physiological Reports, The Journal of Nutritional Physiology and The Journal of Precision Medicine: Health and Disease), please click this link, fill in your name and email address and select 'Research Roundup': <https://www.physoc.org/journals-and-media/membernews>

EDITOR COMMENTS

Reviewing Editor:

Thank you for revising the paper.

REFEREE COMMENTS

Referee #2:

I thank the authors for appropriately addressing all previous comments.